# HURST: Learning Heterogeneity-Adaptive Urban Foundation Models for Spatiotemporal Prediction via Self-Partitional Mixture-of-Spatial-Experts

## Abstract

Urban foundation models (UFMs) are pre-trained spatiotemporal (ST) prediction models with the ability to generalize to different tasks. Such models have the potential to transform urban intelligence by reducing domain-specific models and generalizing to tasks with limited data. However, building effective UFMs is a challenging task due the existence of spatial heterogeneity in ST data, i.e., data distribution and relationship between attributes vary over space. Existing UFMs lack sufficient consideration of this important issue and thus have unsatisfactory performance over spatially heterogeneous urban settings. To address this limitation, this paper proposes **HURST**, a **H**eterogeneity-Adaptive **UR**ban Foundation Model for **S**patio-**T**emporal Prediction, that is capable of capturing the spatial pattern of heterogeneity underlying the urban setting to enhance the UFM's performance. HURST presents two key technical innovations: (1) a self-partitional Mixture-of-Spatial-Experts (MoSE) network that automatically learns to stratify urban areas into partitions, where region-specific expert networks are trained in a hierarchical manner, and (2) an error-guided adaptive spatio-temporal masking strategy that dynamically adjusts masking patterns based on region-specific training feedback. A prompt-tuning strategy is also designed to facilitate the above innovations. Comprehensive experiments on ten datasets from three urban areas of varying sizes show that HURST achieves up to 46.9% performance gain over SOTA baselines.

## 1 Introduction

Spatiotemporal (ST) prediction aims at learning the mapping between past and future observations of attributes over a spatial region. Accurate ST predictions are indispensable for various urban applications such as traffic and mobility prediction (Ji et al., 2023; Jiang et al., 2023; Bao et al., 2020), accident and crime prediction (Mohler et al., 2011; Bogomolov et al., 2014; Jing et al., 2024), etc. Traditional ST prediction models built on deep learning techniques are mostly task-specific, i.e., trained to predict a specific target variable. Such models lack generalizability and are hard to train for tasks with scarce data. Recently, the idea of building Urban Foundation Models (UFMs) emerged as a more attractive solution. Such models are pre-trained on a number of diverse urban datasets to extract comprehensive knowledge and representations of the urban area, thus can be directly applied to additional tasks with limited or no additional training data. A number of UFMs have been proposed and achieved superior performance over task-specific models (Yuan et al., 2018; Dosovitskiy et al., 2021; Liu et al., 2024b; Wang et al., 2019; Chang et al., 2021; Tan et al., 2023; Gao et al., 2022).

Despite the success of UFMs, a key challenge is yet sufficiently addressed. Urban spatiotemporal data typically exhibits strong **heterogeneity**, i.e., the distribution of attribute and their dependencies vary over space and time (Jiang, 2018; An et al.). Such variations can be observed in numerous aspects. For instance, traffic patterns in city centers are markedly different from those in suburban areas, and weekday traffic patterns differ significantly from weekend ones. **ST Heterogeneity creates significant hurdles for UFM's generalizability**, as (1) a predictive model might exhibit drastically

different performance over different ST regions, and (2) different ST attributes may follow distinct generative patterns over space and time, further weakening the cross-task generalizability of UFMs. Unfortunately, existing UFMs have largely overlooked the critical issue of spatiotemporal (ST) heterogeneity. Their architectural designs often lack explicit mechanisms to capture, disentangle, and model the distinct generative patterns specific to different ST contexts. Consequently, these models tend to learn an "averaged" representation that fails to excel in any particular spatial or temporal regime, ultimately resulting in compromised performance across diverse urban scenarios. This limitation is clearly evidenced by the visualized prediction results of various UFMs presented in Fig 8, where their outputs demonstrate inconsistent accuracy and poor adaptability when confronted with varying ST conditions.

Previous techniques on heterogeneity-aware ST prediction models cannot directly solve this issue. Typical ideas include adding heterogeneity-relevant ST input through feature engineering (Yuan et al., 2018), partitioning the study area into homogeneous zones to build region-specific models, and ST transfer learning (Lu et al., 2022b). However, these methods are designed for single tasks rather than pre-trained foundation models. Recently, the Mixture-of-Expert network architecture emerged as a potential solution for data heterogeneity, where multiple expert network are trained simultaneously to handle input with different semantic meanings. However, existing MoE-based ST models (Yu et al., 2024; Wu et al., 2024; Li et al., 2023a) commonly use preset and static expert settings, lacking the ability to adaptively learn task-agnostic heterogeneity-aware representations required by UFMs.

To address the limitations of prior work, this paper presents HURST, a novel **H**eterogeneity-Adaptive **UR**ban Foundation Model for **S**patio-**T**emporal Prediction. HURST features two key innovations to address the heterogeneity challenge: (1) a self-partitional Mixture-of-Spatial-Expert (MoSE) network layer to hierarchically stratify the study area into semantically distinct partitions and simultaneously train corresponding expert networks to capture the pattern of heterogeneity, and (2) an Error-Guided Adaptive Spatiotemporal Masking strategy based on location-specific pretraining feedback to narrow the performance gap. In addition, a new prompt tuning strategy is designed to facilitate the above network design. We conduct comprehensive evaluations for all-for-one, few-shot and zero-shot prediction tasks on ten datasets from three geographic regions. Results show that HURST significantly improves the performance of SOTA UFMs by up to 46.9%. Our key contributions are summarized as follows:

- To the best of our knowledge, HURST is the first work to build an urban foundation model with the spatial heterogeneity challenge explicitly addressed.

- We present a novel self-partitional Mixture-of-Spatial-Experts (MoSE) design to automatically learn hierarchical partitions of the urban area and their corresponding expert networks simultaneously for enhanced representation power.

- We design an Error-Guided Adaptive Masking strategy for model pre-training, which dynamically adjusts mask generation based on location-specific pre-training error to narrow performance gaps over heterogeneous urban regions.

## 2 RELATED WORK

**Spatiotemporal Prediction Models.** The emergence of deep learning has greatly accelerated the development of spatiotemporal prediction techniques. Hetero-ConvLSTM (Yuan et al., 2018) and HintNet (An et al.) solve the heterogeneity of spatial-temporal data. GMAN (Zheng et al., 2020),ASTGCN (Liu et al., 2023a),STGNNs (Kipf & Welling, 2016)and HAGEN (Wang et al., 2022) solve the challenges of spatial-temporal dynamics and heterogeneity through graph neural network. STPP (Yuan et al., 2023) introduces the diffusion model into spatiotemporal prediction. MetaSTC (Xu et al., 2024) provides an interface for acceleration and lightweight traffic prediction model by using meta learning. ST-GFSL (Lu et al., 2022a) proposes to generate non-shared parameters based on node-level meta knowledge. However, these models are designed for specific prediction tasks and lack sufficient generalizability. Models trained on one task (target feature) typically do not perform well on others especially when there is limited data.

**Spaiotemporal and Urban Foundation Models.** Inspired by the Large Language Models and foundation models in natural language processing (Devlin et al., 2019; Howard & Ruder, 2018; Hu et al.,

2022; Brown et al., 2020) and computer vision (Dosovitskiy et al., 2021; Radford et al., 2021; He et al., 2022), researchers have begun exploring the application of pre-trained models in spatiotemporal (ST) domain. For example, UrbanGPT (Li et al., 2024a) and ST-LLM (Liu et al., 2024a) apply LLMs as backbone models for ST prediction, showing strong generalizability. Others build ST foundation models from scratch through pre-training on ST data. STMAE (Sun et al., 2024) and STDMAE (Gao et al., 2024) decouple the relationship between time and space, and learn their representations respectively. GPT-ST (Li et al., 2023b) utilizes hypergraph capsule clustering network to enhance spatial relationship modeling. Methods through pre-training (Sun et al., 2024; Gao et al., 2024; Li et al., 2023b) focus on generating effective spatiotemporal representations to enhance the predictive performance of downstream models. UniST (Yuan et al., 2024) and PromptST (Zhang et al., 2023) have developed complete spatiotemporal pre-training frameworks based on prompt mechanisms. These frameworks enable independent predictions for different spatiotemporal tasks without relying on downstream models. However, these solutions typically ignore the critical challenge of spatiotemporal heterogeneity, which is a key hurdle for model generalizability, resulting in unsatisfactory performance.

**Mixture of Experts (MoE).** The MoE framework enables adaptive pattern assignment to expert modules via hierarchical gating mechanisms (Jacobs et al., 1991; Shazeer et al., 2017). Recent architectures address heterogeneous spatiotemporal dependencies through various implementations, such as self-supervised training (He et al., 2024), transformer-based experts and transfer learning (He et al., 2024), adaptive graph gating for node-level dependency decoupling (Yu et al., 2024), contrastive experts and hierarchical loss re-weighting (Wu et al., 2024), sparse gating with top-k experts for road networks (Li et al., 2023a; Chowdhury et al., 2023; Fedus et al., 2022). However, these approaches primarily utilize static MoE by presetting parameters such as the number of experts and their semantics based on prior experiences. Such design limits model generalizability across tasks as the patterns of heterogeneity are often unknown and need to be adaptively learned. By contrast, our proposed self-partitional Mixture-of-Spatial-Experts (MoSE) architecture simultaneously learn a spatial partitioning scheme and corresponding expert networks to enhances the model's generalizability.

# 3 PRELIMINARIES

## 3.1 DEFINITIONS AND CONCEPTS

**Definition 1: Spatiotemporal region and features.** A spatiotemporal (ST) region is defined as a three-dimensional space $T \times H \times W$, where $T$ denotes uniform time intervals (e.g.,days) and $H \times W$ represents the grid division of the study area on a two-dimensional plane. Spatiotemporal features $X \in R^{T \times H \times W \times C}$ refer to various observations mapped to each grid location of this region (e.g., accidents, crimes, average traffic speed), where $C$ denotes the total number of features.

**Definition 2: Spatiotemporal prediction model.** An ST prediction model $f$ is an approximated mapping from the historical observations of ST feature $X_n$: $X_n^{t-\mathcal{T}:t} \in R^{\mathcal{T} \times H \times W}$ to its value in the next time slot $X_n^t \in R^{1 \times H \times W}$, i.e., $\hat{X}_n^t = f(X_n^{t-\mathcal{T}:t})$.

**Definition 3: Spatiotemporal heterogeneity.** Spatiotemporal heterogeneity refers to the systematic variation of data distributions, statistical properties (e.g., means, variances), or relationships across space and time. Assume spatiotemporal data $X$ is generated by the process $X \sim \Phi$. Spatiotemporal heterogeneity refers to variations in this generation process $\Phi$ across different spatiotemporal locations (Xie et al., 2022). This suggests that the "true" predictive model $f$ for any feature in $X$ also varies over space and time.

## 3.2 PROBLEM FORMULATION

The Heterogeneity-Adaptive Urban Foundation Model Learning Problem is thereby formulated as follows:

- **Input:** Pre-training datasets $D = \{X_1, \dots, X_C\} \in R^{T \times H \times W \times C}$, and test datasets $D' = \{Y_1, \dots, Y_K\} \in R^{T \times H \times W \times K}$
- **Output:** A spatiotemporal prediction model $f_D$ trained on $D$ and fine-tuned on a subset of $D'$ (no fine-tuning set for zero-shot prediction) s.t. $\hat{Y}_i^{t+1} = f_D(Y_i^{t-\mathcal{T}:t}), t \in T, 1 \le i \le K$

- **Objective:** Minimize the prediction error between the ground truth $Y_i^{t+1}$ and the prediction $\hat{Y}_i^{t+1}$ for all the values of $i$ and $t$.
- **Assumptions:** Spatial heterogeneity, as defined above, exists for both training sets $D$ and test sets $D'$. In addition, we also assume that spatiotemporal autocorrelation (i.e., features in nearby grid locations have higher correlations) exists in the datasets, which is a common assumption in similar problems (Xie et al., 2022; Yuan et al., 2018).

## 4 METHODOLOGY

### 4.1 MODEL OVERVIEW

Figure 1 shows the overall architecture of the HURST framework with two stages. In the **Pre-training Stage**, the training set are preprocessed into unified three-dimensional feature tensors and fed into the embedding layers, which extract local features and convert the data into multi-dimensional vector representations. Then, the Error-guided ST masking module generates dynamic masks for reconstruction tasks for pretraining. The Self-Partitioning Mixture-of-Spatial-Experts (MoSE) layer routes embedded features from different sampled regions to corresponding experts, followed by transformer blocks and the projection layer to generate reconstructed input data.

In the **Fine-turning Stage**, the model retains all the components but the mask generation module from the pre-trained model, with a new prompt network introduced to quickly learn mappings between patterns in the new data and the model's knowledge. Detailed design of the model is decribed below.

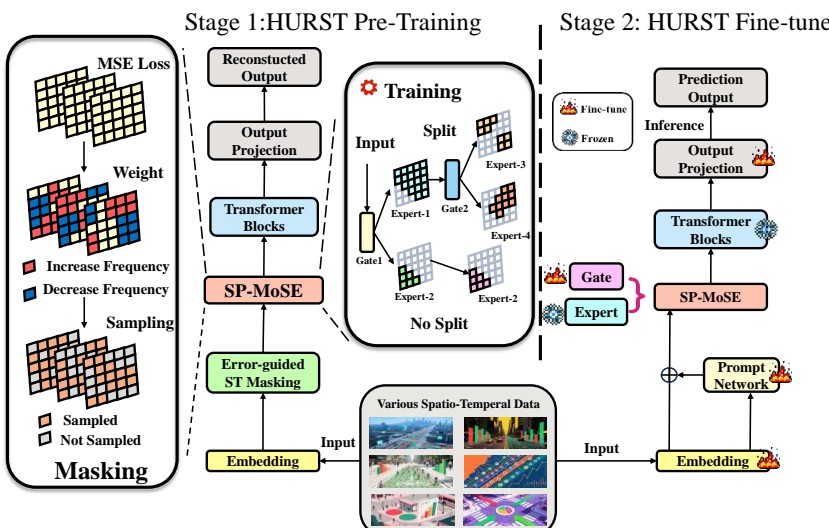

Figure 1: The Overview of HURST

### 4.2 GLOBAL EMBEDDING

This module is designed to generate global representations. Input data $X \in \mathbb{R}^{B \times T \times H \times W}$ undergoes multi-scale feature extraction via cascaded 3D convolution layers. The module then employs a decomposed ST convolution strategy. Specifically, 1D convolution is applied along the temporal axis to capture local temporal patterns, while spatial convolution extracts spatial representations. Finally, shallow and deep features are fused through skip connections to mitigate gradient vanishing. Due to space limit, we illustrate its structure in Fig 5 in the Appendix. The specific formula is as follows:

$$E_{global} = \text{GELU}\left(X + \text{Conv3D}(\text{Conv}_{\text{space}}(\text{Conv}_{\text{time}}(\text{Conv3D}(X))))\right) \qquad (1)$$

### 4.3 Error-guided Spatiotemporal Masking Strategy

In self-supervised learning, the masking strategy plays a crucial role in enhancing the model's understanding of local context. The majority of existing models (Yuan et al., 2024; Zhang et al., 2023; Fang et al., 2024) have chosen random masking strategy or masking designated areas based on external knowledge, which may lead to insufficient training in locations with strong heterogeneity, resulting in poor performance. To address this issue, we propose an error-guided ST masking strategy to guide the model pay more attention to areas with higher reconstruction error and increase their chance of being masked. Specifically, $W \in R^{T \times H \times W}$ represents the masking probability of each ST location, initialized with a uniform distribution. $L_M \in R^{T \times H \times W}$ denotes the reconstruction error at each ST location in the previous epoch of training, which is smoothed by two-dimensional Gaussian kernel at each time slot to enhance spatial coherence, suppress local noise and generate smoother weight distributions. The calculation is done as in Eq 2.

$$L_M^t(i,j) = \sum_{|k| \leq \Omega} \sum_{|l| \leq \Omega} L_M^{(t)}(i+k, j+l) \cdot G_\sigma(k,l), \quad G_\sigma(k,l) = \frac{1}{2\pi\sigma^2} \exp\left(-\frac{k^2+l^2}{2\sigma^2}\right) \quad (2)$$

where $\Omega$ is the Gaussian kernel size typically set to $3\sigma$, $L_M^{(t)} \in \mathbb{R}^{H \times W}$ represents the original loss matrix at time step $t$, $\sigma$ corresponds to the standard deviation of the Gaussian kernel.

Subsequently, the masking weights $W^{new} \in R^{T \times H \times W}$ at each ST location are updated based on $L_M$ and the current weights $W^{old}$ as formulated in Eq 3. Here we only update locations with top and bottom $\beta\%$ of error rates to ensure the stability of the training process. In our emperical evaluation, $\beta = 20$ yields the best performance.

$$W_{(t,x,y)}^{new} = \begin{cases} \min(2 \times W_{(t,x,y)}^{old}, 1) & \text{, if } L_{(x,y)}^t \text{ ranks top-}\beta\% \\ \frac{1}{2} \times W_{(t,x,y)}^{old} & \text{, if } L_{(x,y)}^t \text{ ranks bottom-}\beta\% \\ W_{(t,x,y)}^{old} & \text{, otherwise} \end{cases} \quad (3)$$

Finally, the ST mask $M \in \{0,1\}^{T \times H \times W}$ is generated by an importance-sampling step to pick $r\%$ of valid positions proportionally to $W^{new}$. The sampled positions is set to 0 (for reconstruction), while all the other valid positions remains 1. We evaluate the impact of $r$ in the Experiments section.

### 4.4 Self-Partitional Mixture of Spatial Experts

To further enhance the model's ability to handle spatial heterogeneity, we design a new self-partitional MoSE layer with the goal of learning different expert networks for regions with heterogeneous ST semantics and contexts (e.g., business, residential, urban, rural, etc). Different from prior work, we aim at automatically learning a proper spatial partitioning for experts and their network parameters simultaneously.

The SP-MoSE layer is trained in a hierarchical manner. The gating network first computes weights for an incoming embedding from the previous layer. Then, each location in the embedding is assigned to the best-matched expert based on top-1 selection. Each expert then generates its own part of the embedding representation based on its region-specific parameters, and the final representation is obtained by assembling the outputs of all leaf-node experts. Furthermore, an expert will split into two child experts with a new gating network when its training performance reaches bottleneck (details discussed below).

**Multi-Scale Expert Networks**. Each expert module employs 3D convolutional layers with progressively diminishing receptive fields across hierarchical levels. Lower-level experts focus on global periodic patterns, while higher-level counterparts specialize in localized dynamics. Each non-leaf expert is connected with a trainable gating network to route its outputs to its child experts in the next level. Correspondingly, each expert $Ex_i$'s assigned spatial region is also a subset of its father expert $Ex_{f_i}$'s, creating a increasingly specialized and localized expert.

**Gating Network.** The spatiotemporal gating network generates weights for routing through Eq 4, and the embeddings are routed to the experts with higher weights via the gating network, where $d$ is the current depth in the hierarchical MoSE, $i$ is the serial number of the expert, $j$ is the serial number of the gate, which satisfies the $j$-th gate controls the $2j$-th and the $(2j+1)$-th experts, $Ex_{parent}^i$ denotes the output of the $i$-th expert's father expert, $K$ is the max depth, $\tau_0$ is temperature parameter

controlling entropy of weight distribution.

$$g_d^j = softmax(\frac{Conv3D(Ex_{parent}^{2j})}{\tau_0}) \in R^{T \times H \times W \times 2} \tag{4}$$

**When and How to Split.** Each expert has its own assigned region $R \in \{0,1\}^{T \times H \times W}$ according to the expert weights at the routing gates . We compute the MSE error for each expert in its covered region. If the total loss doesn't decrease for $p$ epochs (patience value) but model training has not stopped, the expert with the highest error is chosen for splitting. Its parameters and those of the corresponding gating module are frozen, and new gates and sub-experts are generated, as depicted in Fig 2.The father expert's receptive field is larger than that of the children experts. To prevent model instability when adding new experts, we initialize the children experts' parameters using the central part of the parent expert's convolution kernel.

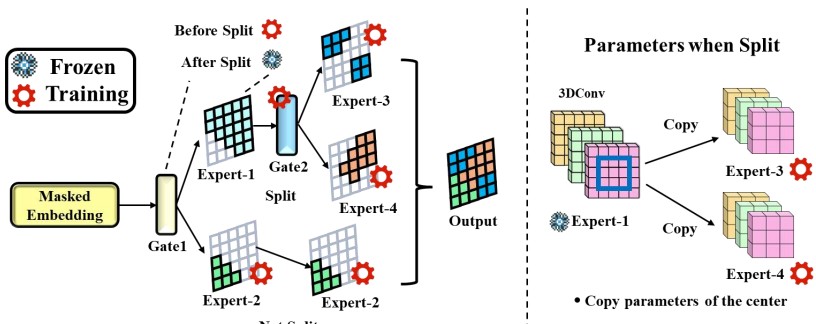

Figure 2: A detailed illustration of the SP-MoSE layer and the expert split process

**Learning Spatial Partitions:** To ensure that each expert is assigned a contiguous and semantically meaningful region in the study area, we incorporate a clustering loss regularization term in model training. This is based on the observation that nearby locations tend to share similar region types and patterns. For each training batch, we calculate the average pair-wise cosine similarity of expert weights between all pairs of locations within a spatial neighborhood window of each other and use this measure as the spatial clustering loss $\mathcal{L}_S$ in model training, as shown in Eq 5,

$$\mathcal{L}_S = -\frac{1}{N} \sum_{(i,j)} \sum_{(m,n) \in \mathcal{N}_{(i,j)}} cos(g_{(i,j)}, g_{(m,n)}) \tag{5}$$

where $\mathcal{N}$ indicates a spatial neighborhood window centered around $(i,j)$, and $N$ represents the size of the window, $g_{(m,n)} \in R^E$ indicates weight of each expert at location $(m,n)$. $\mathcal{L}_S$ encourages the routing gates to assign nearby locations with similar expert weights, thereby increasing their chance of being assigned to the same expert to form spatially contiguous partitions.

$$Ex_{\text{MoSE}} = \sum_{e \in leaf\_experts} \left( \prod_{h \in path(e)} g_h \right) \cdot Ex_e \tag{6}$$

The output $Ex_{\text{MoSE}}$ of the module is obtained by assembling the outputs of leaf experts, i.e., sum of leaf expert outputs weighted by their respective cumulative gating weights through the first level, as shown in Eq 6. This output ultimately passes through the transformer blocks and a linear projection layer to obtain the reconstructed data.

### 4.5 PROMPT NETWORK

Inspired by UniST (Yuan et al., 2024), we utilize a memory-pool-based dual-channel prompt generator to generate spatial features and temporal dependencies for current spatiotemporal few-shot tasks. This generator decouples spatiotemporal attention mechanisms from a scalable memory pool (Weston et al., 2015). Temporal prompts employ a learnable temporal embedding matrix and scaled

dot-product attention to detect periodic patterns and compute temporal weight distributions. Spatial prompts use 2D Convolution to extract local spatial context and improve region-specific pattern capture via channel grouped convolution. An illustration is depicted in Fig 6 in the Appendix.

$$p = [p_s || p_t] \tag{7}$$

where $p_s$ is spatial prompt, $p_t$ is temporal prompt, then concatenated to generate the prompt $p$. Finally, $p + E_{global}$ is fed into the next MoSE layer. How $p_s$ and $p_t$ are calculated is in the Appendix.

## 4.6 MODEL TRAINING

The loss function is composed of three key components. First, MSE loss $\mathcal{L}_{MSE}$ is use to enhance the accuracy of predictions. Second, we incorporate the load balancing loss $\mathcal{L}_{LB}$, which is commonly employed in MoE frameworks (He et al., 2024), to help prevent forming dominantly large partitions with overloaded experts. Third, we include the spatial clustering loss previously introduced to help learn spatially contiguous partitions for each expert. The final loss function is presented as follows,

$$\mathcal{L}_{LB} = \sum_{e=1}^{\mathcal{E}} \frac{Std(Load(e))}{Mean(Load(e))}, Load(e) = \sum_{(i,j)} g_{(i,j)}^e \tag{8}$$

$$\mathcal{L} = \lambda_1 \mathcal{L}_{MSE} + \lambda_2 \mathcal{L}_{LB} + \lambda_3 \mathcal{L}_S \tag{9}$$

where $\mathcal{E}$ denotes the number of leaf experts, $g_{(i,j)}^e$ denotes the weight of expert $e$ at location $(i,j)$. During the training phase, the weights assigned to these three loss components are dynamically adjusted: the weight of the MSE loss component is gradually reduced, while the weights of the load balance loss and spatial clustering loss are progressively increased as expert parameters stabilize.

The pre-training and fine-tuning stages share the same loss function but differ in training strategy. In the pre-training stage, experts in the MoSE layer splits as previously introduced until the model converges or reaches the maximum number of experts. In the fine-tuning stage, the parameters of the experts and the attention parameters of the Transformer blocks are frozen, while other parameters (e.g., prompt network, gating networks) can be fine-tuned. The specific algorithms for the two stages are shown in the Algorithm 2 and Algorithm 3 in Appendix.

## 5 EVALUATION

### 5.1 EXPERIMENTAL SETTING

To rigorously evaluate the performance of the HURST model, we conduct comprehensive experiments on ten distinct urban datasets spanning three geographic regions: New York City (4 datasets), Chicago (4 datasets), and the state of Iowa (2 datasets). The training set, validation set, and test set are temporally partitioned at a ratio of 6:2:2. New York City is divided into $64 \times 64$ grids. Chicago is divided into $64 \times 80$ grids. The state of Iowa is divided into $128 \times 64$ grids. The detailed data description is in the Appendix Table 5, Table 4, Table 6.

**Baselines.** We have carefully selected state-of-the-art models, which can be classified into two distinct categories: single-task-oriented models and pre-trained models. **(1) Single-task models** include Hetero-ConvLSTM (Yuan et al., 2018), ViT (Dosovitskiy et al., 2021), iTransformer (Liu et al., 2024b), MIM (Wang et al., 2019), MAU (Chang et al., 2021), TAU (Tan et al., 2023), SimVP (Gao et al., 2022), TESTAM Lee & Ko (2024) and STARFormer Liu et al. (2023b). For each specific task, a separate model is trained, and subsequently, their performance metrics are reported. **(2) Pre-trained ST models** include PatchTST (Nie et al., 2023), PromptST (Zhang et al., 2023), OpenCity (Li et al., 2024b), and Unist (Yuan et al., 2024). These pre-trained models are trained across multiple datasets and are then employed to complete the corresponding prediction tasks.

**Metrics.** We choose Mean Squared Error (MSE) and Mean Absolute Error (MAE) as evaluation metrics. These two metrics are commonly used in spatiotemporal prediction, where smaller values indicate a more accurate prediction result. All experiments are conducted on a system equipped with a NVIDIA A800 GPU and an Intel Xeon Silver 4310 CPU. The batch size for all experiments is 32. All the reported numbers are the average of five repeated runs. **The full results with standard deviations reported are shown in Appendix A.3.3**.

## 5.2 One-for-All Prediction Results

For single-task models, we trained a dedicated model for each dataset. For pre-trained ST models, we trained a one-for-all model using all the data. Table 1 presents the short-term (one-step) prediction results on 5 datasets from Chicago and Iowa. As shown in the table, our method outperforms baseline models in prediction, with MSE improving by up to 46.9% and MAE by up to 45.2%. The model also achieves higher average performance gain on the Iowa dataset than on the Chicago dataset. This is because the Iowa data has higher heterogeneity, which proves that our model effectively addresses spatiotemporal heterogeneity. Table 2 presents the long-term (seven-step) prediction results, which shows that HURST still maintains excellent performance in long-term prediction. Owing to space limit, results for the New York dataset are presented in Table 15 and Table 16 in the Appendix.

Table 1: One-for-All Short-term Prediction MSE & MAE on Datasets from Chicago and Iowa. C stands for Chicago and I stands for Iowa. **Bold** indicates best, underline indicates second-best.

| Dataset | Taxi Pick(C) | | Taxi Drop(C) | | Traffic Speed(C) | | Share Pick(C) | | Traffic Volume(I) | |
|---|---|---|---|---|---|---|---|---|---|---|
| | MSE | MAE | MSE | MAE | MSE | MAE | MSE | MAE | MSE | MAE |
| HConvLSTM | 0.3048 | 0.2548 | 0.2348 | 0.1985 | 1.4351 | 0.7514 | 15.6792 | 1.4892 | 29.1248 | 4.8745 |
| ViT | 0.2589 | 0.1907 | 0.1840 | 0.0928 | 0.3463 | 0.3158 | 14.4445 | 0.8971 | 27.7283 | 4.9621 |
| ITransformer | 0.2011 | 0.1262 | 0.1808 | 0.0995 | 0.2355 | 0.2135 | 11.0457 | 0.6597 | 6.9610 | 1.2202 |
| SimVP | 0.2026 | 0.1804 | 0.1710 | 0.1261 | 0.4782 | 0.5420 | 10.3842 | 0.6273 | 20.3130 | 3.9968 |
| MIM | 0.2168 | 0.1431 | 0.1879 | 0.1437 | 0.1641 | 0.1715 | 10.9016 | 0.6827 | 9.2633 | 1.3439 |
| MAU | 0.2782 | 0.1912 | 0.1791 | 0.1574 | 0.3027 | 0.2489 | 12.5416 | 1.2547 | 18.1572 | 3.4521 |
| TAU | 0.2512 | 0.1657 | 0.1654 | 0.1412 | 0.2515 | 0.2232 | 13.5419 | 1.0578 | 12.1478 | 2.8941 |
| STID | 0.2017 | 0.0998 | 0.1624 | 0.0741 | 0.1428 | 0.1263 | 12.4874 | 0.8154 | 11.5791 | 3.0157 |
| TESTAM | 0.2354 | 0.2047 | 0.1824 | 0.1866 | 0.2448 | 0.2687 | 13.4571 | 1.2401 | 12.4574 | 2.8743 |
| STAEFromer | 0.2248 | 0.1783 | 0.1674 | 0.1754 | 0.2349 | 0.2474 | 12.5798 | 0.9487 | 12.1547 | 2.8124 |
| PatchTST | 0.4564 | 0.4291 | 0.3096 | 0.3446 | 1.1036 | 0.8350 | 21.2614 | 2.9259 | 12.5479 | 3.1246 |
| Promtst | 0.2116 | 0.2522 | 0.1719 | 0.2112 | 0.3382 | 0.2962 | 16.7459 | 0.7157 | 4.7001 | 1.5895 |
| Opencity | 0.2571 | 0.2318 | 0.2055 | 0.1719 | 0.2176 | 0.2556 | 16.4281 | 1.1087 | 14.2149 | 3.1567 |
| Unist | 0.1938 | 0.0907 | 0.1552 | 0.0608 | 0.1047 | 0.1098 | 12.3674 | 0.7672 | 11.3074 | 2.7604 |
| HURST (ours) | **0.1547** | **0.0497** | **0.1271** | **0.0474** | **0.0796** | **0.0767** | **10.1569** | **0.5958** | **2.4963** | **0.7387** |
| Reduction | 20.1% | 45.2% | 18.1% | 22.1% | 23.9% | 29.9% | 13.5% | 36% | 46.9% | 39.5% |

Hyperparameters: NYC: $\mathcal{L}_s$ loss spatial window $N$=5x5, max. experts $\mathcal{E}$=4, mask rate $r$=0.4, embedding dim. $\mathcal{D}$= 64. Chicago: $N$= 5x5, $\mathcal{E}$=6, $r$=0.4, $\mathcal{D}$=64. Iowa: $N$=7x7, $\mathcal{E}$=4, $r$=0.4, $\mathcal{D}$=128.

## 5.3 Zero-shot and Few-shot Prediction

In practice, ST data may be sparse or missing, so the ability to perform zero-shot and few-shot predictions is vital for evaluating urban generalizable models. Fig 3 presents the prediction results on four datasets of New York City. As shown in the figure, our model surpasses other pre-trained models in zero-shot prediction, demonstrating strong task transferability. Moreover, fine-tuning with just 5%–20% of the data markedly boosts predictive accuracy, highlighting the model's ability to rapidly adapt to new tasks with limited data. The MSE of zero shot prediction increased by up to 52.8%. The rest of results are shown in the Appendix.Notably, all zero-shot and few-shot experiments used the same hyperparameter settings as the one-for-all prediction experiments, ensuring a fair and consistent evaluation framework.

## 5.4 Ablation Study

This subsection explores the importance of the key components in our model. First, we removed each key component one by one and retrained both the pre-trained model and the downstream task model. Then, we compared their performance with that of a model with a complete workflow on the New York dataset. The results are presented in the Table 3, from which we can conclude that each key component of our model plays a crucial role. Specifically, w/o MoSE means we use a linear layer instead of MoSE layer, and w/o Mask means we adopt random masking rather than the proposed error-guided strategy.

Table 2: One-for-All Long-term Prediction MSE & MAE on Datasets from Chicago and Iowa. C stands for Chicago and I stands for Iowa. **Bold** indicates best, underline indicates second-best.

| Dataset | Taxi Pick(C) | | Taxi Drop(C) | | Traffic Speed(C) | | Share Pick(C) | | Traffic Volume(I) | |
| --- | --- | --- | --- | --- | --- | --- | --- | --- | --- | --- |
| | MSE | MAE | MSE | MAE | MSE | MAE | MSE | MAE | MSE | MAE |
| HConvLSTM | 0.3658 | 0.3153 | 0.2818 | 0.2382 | 1.4217 | 0.9017 | 17.8150 | 1.7870 | 30.9498 | 5.0494 |
| ViT | 0.3107 | 0.2268 | 0.2208 | 0.1014 | 0.4156 | 0.3790 | 15.3334 | 1.0765 | 33.2740 | 5.9545 |
| ITransformer | 0.2613 | 0.1534 | 0.2022 | 0.0994 | 0.2826 | 0.2562 | 14.2548 | 0.9916 | 8.3532 | 2.3125 |
| SimVP | 0.2431 | 0.2135 | 0.1752 | 0.1313 | 0.5738 | 0.5504 | 14.4610 | 1.0528 | 21.3756 | 3.7962 |
| MIM | 0.2602 | 0.1747 | 0.2255 | 0.1724 | 0.1969 | 0.2058 | 15.0819 | 0.8192 | 11.1160 | 1.6127 |
| MAU | 0.3338 | 0.2304 | 0.2149 | 0.1889 | 0.3632 | 0.2987 | 15.0499 | 1.5056 | 21.7886 | 4.1425 |
| TAU | 0.3014 | 0.1988 | 0.1985 | 0.1694 | 0.3018 | 0.2678 | 16.2503 | 1.2694 | 14.5774 | 3.4729 |
| STID | 0.2420 | 0.1198 | 0.1949 | 0.0989 | 0.2086 | 0.1916 | 13.9647 | 0.9785 | 13.8949 | 3.6188 |
| TESTAM | 0.2825 | 0.2456 | 0.2189 | 0.1839 | 0.2938 | 0.2324 | 16.1485 | 1.4881 | 14.9489 | 3.1492 |
| STAEFromer | 0.2698 | 0.2140 | 0.2009 | 0.2005 | 0.2819 | 0.2969 | 15.0958 | 1.1384 | 11.5856 | 2.3749 |
| PatchTST | 0.4433 | 0.5578 | 0.4125 | 0.4480 | 1.2347 | 0.8545 | 23.6433 | 3.8037 | 13.3123 | 3.2620 |
| Promtst | 0.2439 | 0.3026 | 0.2063 | 0.2534 | 0.4058 | 0.3554 | 20.0951 | 0.9206 | 5.6401 | 1.6074 |
| Opencity | 0.2785 | 0.2782 | 0.2366 | 0.2063 | 0.2611 | 0.3067 | 19.7137 | 1.3304 | 17.0579 | 3.7880 |
| Unist | 0.2126 | 0.1088 | 0.1662 | 0.0830 | 0.1856 | 0.1818 | 13.8409 | 0.8971 | 10.5689 | 1.4642 |
| HURST (ours) | **0.1627** | **0.0765** | **0.1375** | **0.0703** | **0.1707** | **0.1759** | **13.6970** | **0.8527** | **4.8539** | **1.0351** |
| Reduction | 23.5% | 29.7% | 17.3% | 15.3% | 8.0% | 3.2% | 1.0% | 4.9% | 13.9% | 29.5% |

Hyperparameters: NYC: $\mathcal{L}_s$ loss spatial window $N$=5x5, max. experts $\mathcal{E}$=4, mask rate $r$=0.4, embedding dim. $\mathcal{D}$= 64. Chicago: $N$= 5x5, $\mathcal{E}$=6, $r$=0.4, $\mathcal{D}$=64. Iowa: $N$=7x7, $\mathcal{E}$=4, $r$=0.4, $\mathcal{D}$=128.

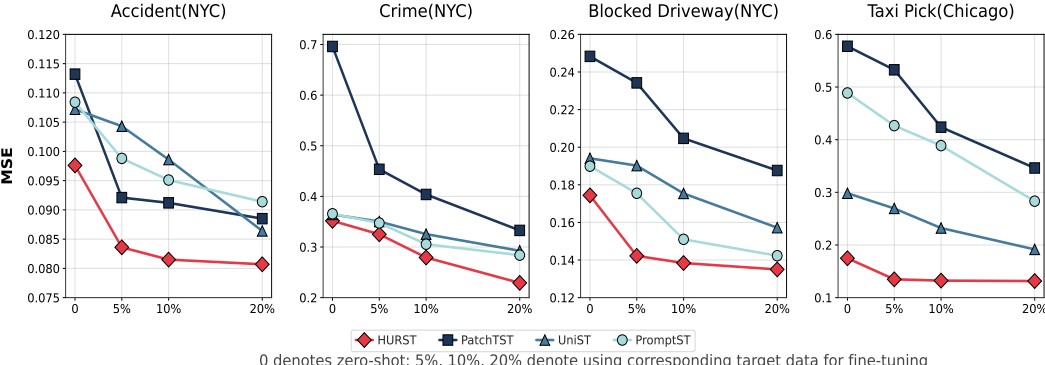

0 denotes zero-shot; 5%, 10%, 20% denote using corresponding target data for fine-tuning

Figure 3: Zero-shot and Few-shot Prediction Results

## 5.5 CASE STUDY

In this section, we visualize the distribution of experts in three real regions and try to interpret their semantics as shown in the Fig 4. Specifically, in New York City, the green zone is mainly composed of areas such as Manhattan, Brooklyn, Queens, etc., with higher population density and rich samples. The yellow expert mainly covers coastal areas and less populated regions, while the data distribution in the blue expert's area is the sparsest. For Iowa, the yellow area is mainly composed of cities and towns with higher traffic flows, and the blue expert covers rural roads with limited traffic. For Chicago, the yellow expert covers the central urban area, while the red expert covers hotspots in downtown and busy routes near the O'Hare airport.Owing to space limitations, the analysis regarding the justification of expert division is provided in the Appendix A.3.3.

## 6 CONCLUSION

This paper proposed HURST, a heterogeneity-adaptive urban foundation model for spatio-temporal prediction. To the best of our knowledge, this was the first urban foundation model that explicitly addressed the key challenge of spatial heterogeneity. Two technical innovations were presented, including (1) a Self-partitional Mixture-of-Spatial-Experts, which learns spatial partitioning of the

Table 3: Ablation Study on Four New York Datasets (MSE & MAE)

| Method | Accident | | Crime | | Illegal Parking | | Blocked Driveway | |
|---|---|---|---|---|---|---|---|---|
| | MSE | MAE | MSE | MAE | MSE | MAE | MSE | MAE |
| HURST | **0.0807** | **0.0933** | **0.2291** | **0.1171** | **0.2778** | **0.1457** | **0.1350** | **0.1041** |
| w/o Prompt | 0.0945 | 0.1121 | 0.3777 | 0.2443 | 0.5218 | 0.2867 | 0.2190 | 0.1838 |
| w/o MoSE | 0.0962 | 0.1355 | 0.3005 | 0.1142 | 0.3135 | 0.1402 | 0.1624 | 0.2033 |
| w/o $\mathcal{L}_s$ | 0.1059 | 0.1864 | 0.6283 | 0.3414 | 0.4853 | 0.3503 | 0.2306 | 0.2508 |
| w/o Mask | 0.0828 | 0.0981 | 0.3379 | 0.1500 | 0.3034 | 0.1744 | 0.1551 | 0.1274 |

**Hyperparameters**: Spatial window $N = 5 \times 5$, experts $\mathcal{E} = 4$, mask rate $r = 0.4$, embedding dim. $\mathcal{D} = 64$.

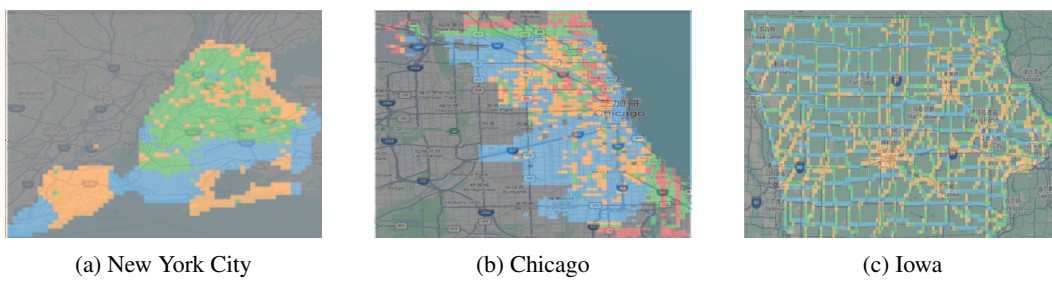

(a) New York City  (b) Chicago  (c) Iowa

Figure 4: Expert Distribution in Three Geographic Regions

study area and expert networks simultaneously, and (2) a error-guided spatiotemporal masking strategy for pretraining. Evaluations on ten datasets from three regions validated the superior performance of HURST over SOTA baselines.

## 7 ETHICS STATEMENT

This study proposes a heterogeneity-adaptive urban foundation model (HURST) for spatiotemporal prediction, strictly adhering to the ICLR Code of Ethics in data usage, model design, and experimental evaluation. All datasets used in this research are publicly available anonymized urban data (e.g., traffic, accident, and public service request data from New York City, Chicago, and Iowa), containing no personally identifiable information. During data preprocessing, all sensitive attributes (such as race, gender, and age) were removed, and only grid-level aggregated statistical features were utilized to prevent privacy leakage risks at the source. The experimental design follows the principle of minimal data usage, extracting only essential spatiotemporal features for model training. This study has no conflicts of interest, and all experiments were conducted in a compliant high-performance computing environment without involving human subjects or sensitive operations.

## 8 REPRODUCIBILITY STATEMENT

To support the reproducibility of this study, we have deposited the relevant code in an anonymized repository( https://anonymous.4open.science/r/hurst-BE8F/). Researchers can use this code to verify all experimental results reported in the paper and build upon this work for further investigations. We will maintain the repository actively and provide timely technical support.

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

## A  APPENDIX

### A.1  ADDITIONAL DETAILS OF HURST

Due to space limitations, the figures for 4.2 Global Embedding and 4.5 Prompt Network are presented in this section. In global embedding module, we extract temporal, spatial, and spatiotemporal information from the data as embeddings, providing subsequent modules with comprehensive representations, as shown in Fig 5. As for prompt network, illustrated in Fig 6, it learns temporal and spatial representations for spatiotemporal tasks. For different tasks, it matches relevant information from the memory pool to generate effective task representations, enhancing prediction accuracy. Spatial prompt $p_s$ and temporal prompt $p_t$ are calculated by Eq 11 and Eq 10.

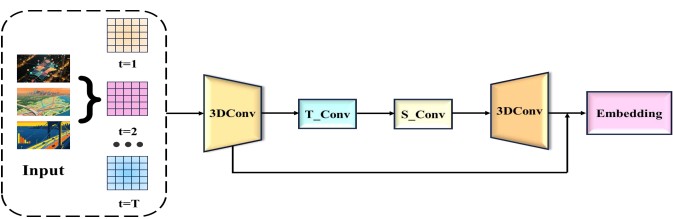

Figure 5: Global Embedding Module

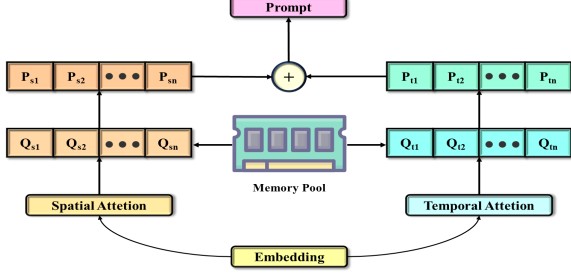

Figure 6: Prompt Network

$$Q_s = Conv2D(E_{global}), Q_t = Attention(E_{global}) \tag{10}$$

$$p_t = \sum_{j=1}^{N_k} \text{softmax}(Q_t K_j^\top / \tau_1) \cdot V_j, p_s = \sum_{j=1}^{N_k} \text{softmax}(Q_s K_j^\top / \tau_1) \cdot V_j \qquad (11)$$

where $\tau_1$ denotes temperature parameters, $Q$, $K$, and $V$ are respectively query,key and value.

## A.2 ALGORITHM PROCESS

Algorithm.1 illustrates the complete process of Error-guided Spatiotemporal Masking Strategy. Algorithm.2 illustrates the process of pre-training stage of HURST. Algorithm.3 illustrates the process of the fine-tuning stage of HURST.

---

**Algorithm 1** Error-guided Spatiotemporal Masking Strategy

---

**Input:** $E \in \mathbb{R}^{T \times H \times W}$: Embeddings
  $r$: Base mask ratio
  $\beta$: Top/bottom selection ratio
**Output:** $M_{\text{new}}$: Updated mask
  $E_{\text{masked}}$: Masked embeddings
  Initialize $W \leftarrow \mathbf{1}^{T \times H \times W}$
  **for** $t = 1$ to $T$ **do**
    $L_{\text{smooth}}^t \leftarrow \text{GaussianFilter}(L^t)$
    Update weights:

$$W_{(t,x,y)}^{new} = \begin{cases} \min(2 \times W_{(t,x,y)}^{old}, 1) & \text{if } L_{(x,y)}^t \text{ranks Top-}\beta \\ \frac{1}{2} \times W_{(t,x,y)}^{old} & \text{if } L_{(x,y)}^t \text{ranks Bottom-}\beta \\ W_{(t,x,y)}^{dlo} & \text{otherwise} \end{cases}$$

    $S^t \leftarrow \text{MultinomialSample}(W^t, r)$
    Generate mask:

$$M_{\text{new}}^t(x, y) = \begin{cases} 0 & \text{if } (x, y) \notin \text{ValidGrid} \vee (x, y) \in S^t \\ 1 & \text{otherwise} \end{cases}$$

    $E_{masked} = E \cdot M_{\text{new}}$
  **end for**

---

---

**Algorithm 2** Pre-Training Strategy

---

**Input:** Pre-training datasets $D = \{X_1, \ldots, X_C\} \in R^{T \times H \times W \times C}$
  $P$:patience
**Output:** $\theta$: Model parameters
  Initialize $\theta$,dataset importance list $Imp_D \in R^C$
  **for** $epoch = 1$ to $N$ **do**
    Sample a dataset X for each batch according to $Imp_D$
    Generate reconstructed $\hat{X}$
    Compute loss $\mathcal{L}$ by Eq.9
    Compute MSE loss for each leaf-expert
    **if** the total loss $\mathcal{L}$ hasn't decreased over $P$ epochs **then**
      Split the expert with Maximum mse loss
      Frozen the paremeters of the splited expert and its parent gate
      Add two sub-experts
      Copy the center parameters of the parent expert and assign them to the child experts
    **end if**
    update $\theta$
  **end for**

---

---

**Algorithm 3** Fine-tuning Strategy

---

**Input:** $X_{tar}$: Target Dataset
 $\theta$: Model parameters from pre-training stage
**Output:** $\hat{y_{tar}}$:Prediction result of target dataset
 $\phi$:parameters of prediction model
 Initialize:model parameters $\phi$
 Load parameters $\theta$ from pre-training stage
 Frozen the parameters of experts and transformer attention layers in $\theta$
 **for** $epoch = 1$ to $N$ **do**
  Generate $\hat{y_{tar}} = f_\phi(X_{tar})$
  compute Loss $\mathcal{L}(\hat{y_{tar}}, y_{tar})$
  update parameters $\phi$
 **end for**

---

### A.3 EXPERIMENTAL DETAILS

#### A.3.1 DATA.

We conducted extensive experiments on the datasets of New York[1], Chicago[2], and Iowa[3], with all dataset details presented in the Table.4, Table.5, Table.6 respectively.

Table 4: Description of NYC Urban Datasets (2019-2021) on 64×64 Grid

| Dataset | Max | Min | Mean | Std |
|---|---|---|---|---|
| Illegal Parking | 83 | 0 | 0.16 | 0.73 |
| Accident | 13 | 0 | 0.09 | 0.39 |
| Plumbing | 74 | 0 | 0.03 | 0.28 |
| Bulky Item Collection | 20 | 0 | 0.18 | 0.68 |
| Street Condition | 108 | 0 | 0.04 | 0.3 |
| Blocked Driveway | 29 | 0 | 0.09 | 0.45 |

Table 5: Description of Chicago Urban Metrics (2019–2021) on 64×80 Grid

| Dataset | Max | Min | Mean | Std |
|---|---|---|---|---|
| Taxi Pickups | 171.1250 | 0 | 0.1668 | 2.4783 |
| Taxi Dropoffs | 118.5833 | 0 | 0.1794 | 2.1817 |
| Traffic Speed | 12.8222 | 0 | 0.1626 | 0.4905 |
| Traffic Volume | 0.2816 | 0 | 0.0034 | 0.0106 |
| Shared Vehicle Pickups | 482.4583 | 0 | 1.4400 | 7.9722 |
| Shared Vehicle Dropoffs | 498.9583 | 0 | 1.4208 | 8.1162 |

Table 6: Description of Iowa Traffic Data (2016-2018) on 128×64 Grid

| Dataset | Max | Min | Mean | Std |
|---|---|---|---|---|
| Average Traffic Speed | 89.84 | 0 | 19.13 | 28.01 |
| Normal Vehicle Traffic Volume | 80.62 | 0 | 2.05 | 3.21 |
| Truck Traffic Volume | 79.40 | 0 | 1.03 | 1.70 |
| Accident Count | 22.00 | 0 | 0.02 | 0.20 |

---

[1] https://opendata.cityofnewyork.us/
[2] https://data.cityofchicago.org/Transportation/Traffic-Crashes-Crashes/85ca-t3if
[3] https://iowadot.gov/

### A.3.2 PARAMETERS.

The proposed method is trained by minimizing the loss in Eq.9 by using the Adam optimizer with settings $\alpha = 10^{-3}, \beta_1 = 0.9, \beta_2 = 0.999, \epsilon = e \cdot 10^{-8}, \lambda_1 = 10, \lambda_2 = 0.01, \lambda_3 = 0.001, \beta = 0.2$. Table 7 summarizes the parameter counts and per-epoch training times of various UFMs. From the results in the table, it can be concluded that the HURST model does not improve its performance by increasing the number of parameters. It has fewer parameters than Unist and PatchTST but comparable training time cost.

Table 7: Model Parameters and Training Time Comparison

| Model | Parameters Count | Per-Epoch Training Time |
|---|---|---|
| HURST | 9.7M | 13s |
| UNIST | 10.8M | 4s |
| PATCHTST | 11.6M | 4s |
| PromptST | 3.5M | 243s |
| OpenCity | 6.2M | 14min |

Fig 7 in the Appendix shows how $\mathcal{L}_c$ loss window size, number of experts, embedding layer dimensions, and mask rate affect model performance. An insufficient number of experts limits the learning of heterogeneous information. Similarly, a window size that's either too large or too small can compromise the model's ability to capture heterogeneity, resulting in degraded performance. In a similar vein, a low mask rate renders learning too facile, precluding performance enhancement, while an excessive mask rate triggers data scarcity, which also undermines performance.

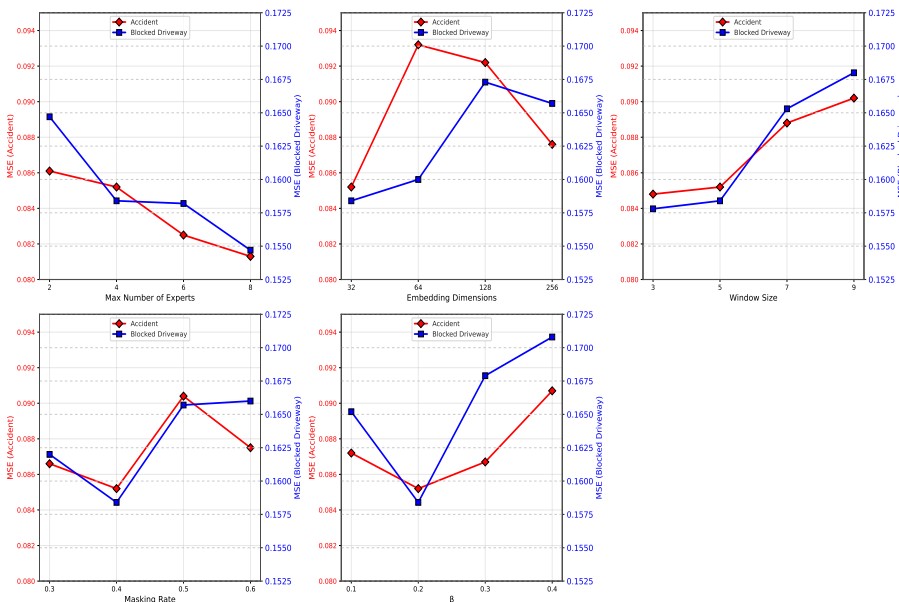

Figure 7: Impact of Parameters on Experimental Results

### A.3.3 ADDITIONAL RESULTS.

In addition to the aggregate MSE and MAE reported in the Sec.5, we further compute the mean and variance of the prediction errors across all the spatial grid cells for each UFM on the Chicago Share-Pick dataset (the same dataset visualized in Fig 8 of the case study). The results are summarized in the Table 8. It can be concluded that HURST, after explicitly considering the heterogeneity of spatiotemporal data, not only has a smaller mean error but also achieves a more consistent performance over different spatial locations compared to UFMs that do not explicitly consider heterogeneity.

Table 8: Model Performance Comparison (Mean and Variance)

| Model | Mean | Variance |
|---|---|---|
| **HURST** | **1.26** | **12.55** |
| Unist | 2.42 | 20.13 |
| PatchTST | 2.08 | 14.48 |
| PromptST | 1.61 | 18.00 |

**Expert Distribution**.In the ablation study, we remove the SP-MoSE module and replace it with a regular linear layer, resulting in a significant increase in the error between the predicted and true values. This indicates that the module has indeed played a key role in improving prediction performance. Meanwhile, taking the Share-Pickup dataset in Chicago as an example, the statistical results of the dataset in different expert regions(colored as show in Fig 4) are shown in the Tabel 9. The mean values of the yellow and red experts' areas (urban hotspots) are indeed higher than those of the green and blue (non-hotspot) areas, which confirms the rationality of the expert distribution. The yellow and red areas mainly include high traffic areas, such as parks (such as Columbus Park Refectory,Altitude Chicago Trampoline Park ,etc.),markets(such as Costco Wholesale, The Buyers Flea Market,etc), United Center and University of Illinois Chicago,while the other two areas are mostly composed of areas with lower traffic flow like residential areas.Meanwhile,the MSE and MAE across different expert regions (color-coded in Fig 4) are summarized in the Tabel 10,which shows that the hotspot regions (red/yellow) exhibit higher errors than non-hotspot regions (blue/green).This occurs because hotspot regions experience frequent unexpected events, which create irregular patterns that challenge model learning. Conversely, non-hotspot regions benefit from more regular patterns, resulting in stabler model performance.

Table 9: Mean and Variance of Data in Each Expert Region

| Color | Mean | Variance |
|---|---|---|
| Yellow | 3.16 | 50.31 |
| Red | 3.15 | 61.87 |
| Blue | 1.96 | 32.50 |
| Green | 2.26 | 79.24 |

Table 10: HURST Performance in Each Expert Region

| Color | Mean | Variance | MSE | MAE |
|---|---|---|---|---|
| Yellow | 3.16 | 50.31 | 16.95 | 1.17 |
| Red | 3.15 | 61.87 | 22.01 | 1.13 |
| Blue | 1.96 | 32.50 | 20.83 | 0.86 |
| Green | 2.26 | 79.24 | 7.47 | 0.68 |

**Influence of Spatio-temporal Heterogeneity on Model Performance**.Spatio-temporal heterogeneity represents a fundamental challenge in urban data modeling, referring to systematic variations in data distributions, statistical properties(e.g., means, variances),and predictive relationships across different geographical locations and temporal periods.To quantify this heterogeneity in our experimental datasets,we compute the Coefficient of Variation (CV) across both spatial and temporal dimensions as shown in Table 11.As demonstrated in our analysis, the Iowa Traffic Speed dataset exhibits the most pronounced heterogeneity, with significantly high CV values in both spatial and temporal dimensions,underscoring the necessity of specialized approaches to handle such distributional variations. We further present each method's (1) standard deviation of prediction MSE over all the grid locations on the three datasets, (2) performance gain (MSE reduction in %) over each dataset's respective historical average. Results in Table 12 demonstrate that baseline models have elevated standard deviations of MSE and less performance gain over historical average under conditions of heightened heterogeneity. By contrast, our HURST model achieves an increase in performance improvement with increasing data heterogeneity on three datasets. Although promptST has also shown such a trend, its standard deviation is greater than patchtst, indicating that its performance

Table 11: Coefficient of Variation (CV) Comparison on Three Geographic Regions Datasets

| Dataset | Average Spatial CV | Average Temporal CV |
|---|---|---|
| NYC-Crime | 5.48 | 0.22 |
| Chicago-Sharepickup | 6.51 | 0.12 |
| Iowa-Traffic Speed | 6.03 | 0.63 |

is not stable enough.This clearly proves the limitations of related work in handling spatiotemporal heterogeneity and the advantages of our proposed model.

Table 12: Performance Comparison on Three Datasets

| NYC-Crime Dataset | | | |
|---|---|---|---|
| **Model** | **Mean of MSE** | **Std of MSE** | **Performance Gain** |
| **HURST** | **0.29** | **0.11** | **57.5%** |
| Unist | 0.30 | 0.26 | 54.5% |
| PatchTST | 0.39 | 0.13 | 43.9% |
| PromptST | 0.61 | 1.07 | 7.6% |

| Chicago-SharePickup Dataset | | | |
|---|---|---|---|
| **Model** | **Mean of MSE** | **Std of MSE** | **Performance Gain** |
| **HURST** | **1.26** | **0.31** | **61.5%** |
| Unist | 2.42 | 4.49 | 53.1% |
| PatchTST | 2.08 | 3.81 | 19.4% |
| PromptST | 1.61 | 4.24 | 36.5% |

| Iowa-Traffic Volume Dataset | | | |
|---|---|---|---|
| **Model** | **Mean of MSE** | **Std of MSE** | **Performance Gain** |
| **HURST** | **2.53** | **0.27** | **82.0%** |
| Unist | 10.72 | 19.04 | 18.4% |
| PatchTST | 12.67 | 1.28 | 9.5% |
| PromptST | 4.73 | 4.07 | 66.1% |

Table 15 and Table 16 show the one-for all prediciton results on four datasets in New York City. In New York, our model's one-for-all and zero-shot prediction accuracies improved by up to 11.5% and 13.6%, respectively, outperforming baselines significantly. Table 19 and Table 20 shows the zero-shot prediction results on four Chicago datasets. The visualization results on the Chicago Sharepick dataset are shown in Figure 8. It can be seen that the model's predictions are meaningful and not merely a reduction in metrics. All the reported numbers are the average of five repeated runs. The standard deviations of the experimental results are presented in the following table

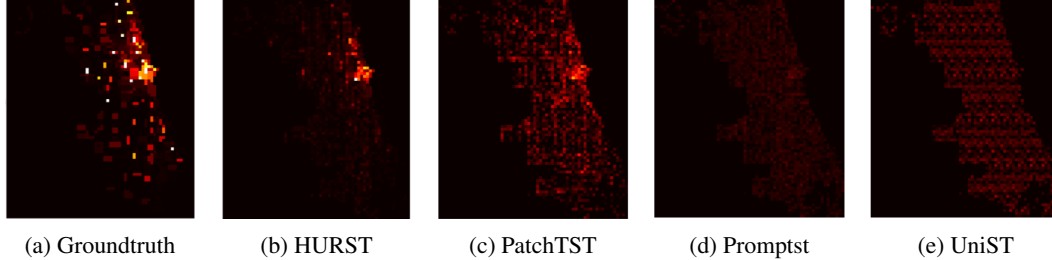

(a) Groundtruth     (b) HURST     (c) PatchTST     (d) Promptst     (e) UniST

Figure 8: Comparison of Visual Prediction Results on Chicago Share Pick

Table 13: One-for-All Short-term Prediction Performance on Chicago Taxi Datasets (Pickup & Dropoff). **Bold** indicates best, underline indicates second-best.

| Dataset | Taxi Pickup | | Taxi Dropoff | | Traffic Speed | |
|---|---|---|---|---|---|---|
| | MSE | MAE | MSE | MAE | MSE | MAE |
| HConvLSTM | 0.3048±0.0244 | 0.2548±0.0204 | 0.2348±0.0188 | 0.1985±0.0159 | 1.4351±0.1284 | 0.7514±0.0582 |
| ViT | 0.2589±0.0207 | 0.1907±0.0153 | 0.1840±0.0147 | 0.0928±0.0074 | 0.3463±0.0217 | 0.3158±0.0185 |
| ITransformer | 0.2011±0.0161 | 0.1262±0.0101 | 0.1808±0.0145 | 0.0995±0.0080 | 0.2355±0.0153 | 0.2135±0.0128 |
| SimVP | 0.2026±0.0170 | 0.1804±0.0144 | 0.1710±0.0137 | 0.1261±0.0101 | 0.4782±0.0286 | 0.5420±0.0321 |
| MIM | 0.2168±0.0173 | 0.1431±0.0114 | 0.1879±0.0150 | 0.1437±0.0115 | 0.1641±0.0098 | 0.1715±0.0103 |
| MAU | 0.2782±0.0223 | 0.1912±0.0153 | 0.1791±0.0143 | 0.1574±0.0126 | 0.3027±0.0181 | 0.2489±0.0149 |
| TAU | 0.2512±0.0201 | 0.1657±0.0133 | 0.1654±0.0132 | 0.1412±0.0113 | 0.2515±0.0151 | 0.2232±0.0134 |
| STID | 0.2017±0.0161 | 0.0998±0.0080 | 0.1624±0.0130 | 0.0741±0.0059 | 0.1428±0.0086 | 0.1263±0.0076 |
| TESTAM | 0.2354±0.0283 | 0.2047±0.0124 | 0.1824±0.0174 | 0.1866±0.0084 | 0.2448±0.0104 | 0.2687±0.0026 |
| STAEFromer | 0.2248±0.0234 | 0.1783±0.0114 | 0.1674±0.0147 | 0.1754±0.0075 | 0.2349±0.0098 | 0.2474±0.0022 |
| PatchTST | 0.4564±0.0456 | 0.4291±0.0429 | 0.3096±0.0309 | 0.3446±0.0345 | 1.1036±0.0883 | 0.8350±0.0501 |
| Promtst | 0.2116±0.0190 | 0.2522±0.0252 | 0.1719±0.0155 | 0.2112±0.0211 | 0.3382±0.0203 | 0.2962±0.0178 |
| Opencity | 0.2571±0.0206 | 0.2318±0.0185 | 0.2055±0.0164 | 0.1719±0.0138 | 0.2176±0.0131 | 0.2556±0.0153 |
| Unist | 0.1938±0.0078 | 0.0907±0.0059 | 0.1552±0.0062 | 0.0608±0.0049 | 0.1047±0.0078 | 0.1098±0.0059 |
| HURST (ours) | **0.1547±0.0032** | **0.0497±0.0028** | **0.1271±0.0054** | **0.0474±0.0011** | **0.0796±0.0062** | **0.0767±0.0028** |
| Reduction | 20.1% | 45.2% | 18.1% | 22.1% | 23.9% | 29.9% |

Table 14: One-for-All Short-term Prediction Performance on Iowa Traffic Datasets (Speed & Volume) and Chicago Share Pickup.

| Dataset | Traffic Speed(I) | | Traffic Volume(I) | | Share Pickup(C) | |
|---|---|---|---|---|---|---|
| | MSE | MAE | MSE | MAE | MSE | MAE |
| HConvLSTM | 32.4519±2.5962 | 5.3216±0.4257 | 29.1248±2.3300 | 4.8745±0.3899 | 15.6792±1.2543 | 1.4892±0.1097 |
| ViT | 29.4692±2.3575 | 3.2645±0.2612 | 27.7283±2.2183 | 4.9621±0.3970 | 14.4445±0.8676 | 0.8971±0.0349 |
| ITransformer | 13.1727±1.0538 | 1.7538±0.1403 | 6.9609±0.5569 | 1.2202±0.0976 | 11.0457±0.6627 | 0.6597±0.0281 |
| SimVP | 13.4500±1.0760 | 2.1983±0.1759 | 20.3130±1.6250 | 3.9968±0.3197 | 10.3842±0.5192 | 0.6273±0.0257 |
| MIM | 14.1249±1.1300 | 3.1487±0.2519 | 9.2633±0.7411 | 1.3439±0.1075 | 10.9016±0.6541 | 0.6827±0.0273 |
| MAU | 16.7891±1.3431 | 3.2149±0.2572 | 18.1572±1.4526 | 3.4521±0.2762 | 12.5416±0.7525 | 1.2547±0.0502 |
| TAU | 11.2146±0.8972 | 2.8944±0.2316 | 12.1478±0.9718 | 2.8941±0.2315 | 13.5419±0.8125 | 1.0578±0.0423 |
| STID | 11.2487±0.8999 | 2.7642±0.2211 | 11.5791±0.9263 | 3.0157±0.2413 | 12.4874±0.7492 | 0.8154±0.0326 |
| TESTAM | 11.0547±0.7864 | 2.5157±0.2140 | 12.4574±0.9846 | 2.8743±0.2224 | 13.4571±0.8452 | 1.2401±0.0264 |
| STAEFromer | 12.5878±0.8453 | 2.6871±0.2235 | 12.1547±0.9642 | 2.8124±0.2147 | 12.5798±0.7419 | 0.9487±0.0211 |
| PatchTST | 20.5894±1.6472 | 4.5781±0.3662 | 12.5479±1.0038 | 3.1246±0.2500 | 21.2614±1.7009 | 2.9259±0.1755 |
| Promtst | 19.4081±1.5526 | 4.2186±0.3375 | 4.7001±0.3760 | 1.5895±0.1272 | 16.7459±1.0047 | 0.7157±0.0286 |
| Opencity | 15.3168±1.2253 | 2.2135±0.1771 | 14.2149±1.1372 | 3.1567±0.2525 | 16.4281±0.9857 | 1.1087±0.0443 |
| Unist | 10.5242±0.5154 | 2.6214±0.1263 | 11.3074±0.9046 | 2.7604±0.2208 | 12.3674±0.5154 | 0.7672±0.0126 |
| HURST (ours) | **9.1021±0.4362** | **1.1219±0.1412** | **2.4963±0.1997** | **0.7387±0.0591** | **10.1569±0.4641** | **0.5958±0.0112** |
| Reduction | 13.5% | 36.0% | 46.9% | 39.5% | 2.2% | 5.0% |

Table 15: One-for-All Short-term Prediction Performance on NYC Accident & Crime Datasets (MSE & MAE).

| Dataset | Accident | | Crime | |
|---|---|---|---|---|
| | MSE | MAE | MSE | MAE |
| HConvLSTM | 0.1087±0.0087 | 0.1005±0.0080 | 0.3256±0.0260 | 0.2571±0.0206 |
| ViT | 0.1014±0.0081 | 0.1159±0.0093 | 0.3130±0.0250 | 0.1820±0.0146 |
| ITransformer | 0.0882±0.0071 | 0.0884±0.0071 | 0.3124±0.0250 | 0.1775±0.0142 |
| SimVP | 0.1011±0.0081 | 0.1278±0.0102 | 0.3137±0.0251 | 0.2034±0.0163 |
| MIM | 0.0901±0.0072 | 0.0891±0.0071 | 0.3406±0.0272 | 0.1511±0.0121 |
| MAU | 0.1014±0.0081 | 0.1021±0.0082 | 0.3845±0.0308 | 0.2014±0.0161 |
| TAU | 0.0989±0.0079 | 0.0916±0.0073 | 0.3154±0.0252 | 0.1647±0.0132 |
| STID | 0.0981±0.0078 | 0.1069±0.0086 | 0.4649±0.0372 | 0.2724±0.0218 |
| TESTAM | 0.1084±0.0088 | 0.0942±0.0086 | 0.2989±0.0351 | 0.2152±0.0283 |
| STAEFromer | 0.0951±0.0083 | 0.1065±0.0081 | 0.3089±0.0387 | 0.1891±0.0344 |
| PatchTST | 0.1114±0.0089 | 0.1441±0.0115 | 0.3653±0.0292 | 0.3515±0.0281 |
| Promtst | 0.1062±0.0085 | 0.1324±0.0106 | 0.6167±0.0493 | 0.5645±0.0452 |
| Opencity | 0.1107±0.0089 | 0.1037±0.0083 | 0.3687±0.0295 | 0.2634±0.0211 |
| Unist | 0.0979±0.0078 | 0.1047±0.0084 | 0.3027±0.0242 | 0.1678±0.0134 |
| HURST (Ours) | **0.0852±0.0068** | **0.0872±0.0070** | **0.2812±0.0225** | **0.1393±0.0111** |
| Reduction | 3.4% | 1.3% | 7.1% | 7.8% |

Table 16: One-for-All Short-term Prediction Performance on NYC Illegal Parking & Blocked Driveway Datasets (MSE & MAE).

| Dataset | Illegal Parking | | Blocked Driveway | |
|---|---|---|---|---|
| | MSE | MAE | MSE | MAE |
| HConvLSTM | 0.3518±0.0281 | 0.1852±0.0148 | 0.1958±0.0157 | 0.1671±0.0134 |
| ViT | 0.3352±0.0268 | 0.1962±0.0157 | 0.1850±0.0148 | 0.1547±0.0124 |
| ITransformer | 0.3381±0.0270 | 0.2058±0.0165 | 0.1848±0.0148 | 0.1267±0.0101 |
| SimVP | 0.3358±0.0269 | 0.1895±0.0152 | 0.1878±0.0150 | 0.1666±0.0133 |
| MIM | 0.3226±0.0258 | 0.1964±0.0157 | 0.1664±0.0133 | 0.1334±0.0107 |
| MAU | 0.3697±0.0296 | 0.2254±0.0180 | 0.2154±0.0172 | 0.1784±0.0143 |
| TAU | 0.3451±0.0276 | 0.2124±0.0170 | 0.2033±0.0163 | 0.1642±0.0131 |
| STID | 0.3172±0.0254 | 0.2136±0.0171 | 0.1722±0.0138 | 0.1425±0.0114 |
| TESTAM | 0.3248±0.0274 | 0.1935±0.0196 | 0.1845±0.0165 | 0.1819±0.0132 |
| STAEFromer | 0.3321±0.0288 | 0.2045±0.0247 | 0.1749±0.0172 | 0.1768±0.0121 |
| PatchTST | 0.3278±0.0262 | 0.3049±0.0244 | 0.1711±0.0137 | 0.1977±0.0158 |
| Promtst | 0.4569±0.0366 | 0.4377±0.0350 | 0.2234±0.0179 | 0.2751±0.0220 |
| Opencity | 0.3492±0.0279 | 0.3517±0.0281 | 0.2358±0.0189 | 0.2040±0.0163 |
| Unist | 0.2904±0.0232 | 0.1844±0.0148 | 0.1627±0.0130 | 0.1590±0.0127 |
| HURST (Ours) | **0.2824±0.0226** | **0.1629±0.0130** | **0.1584±0.0127** | **0.1211±0.0097** |
| Reduction | 2.7% | 11.7% | 2.6% | 4.4% |

Table 17: Zero-shot Performance on Public Safety Datasets (Accident & Crime).

| Dataset | Accident | | Crime | |
|---|---|---|---|---|
| | MSE | MAE | MSE | MAE |
| PatchTST | 0.1132±0.0091 | 0.1802±0.0144 | 0.6961±0.0557 | 0.5801±0.0464 |
| Promtst | 0.1084±0.0087 | 0.0899±0.0072 | 0.3652±0.0292 | 0.2438±0.0195 |
| Unist | 0.1072±0.0086 | 0.1076±0.0086 | 0.3643±0.0291 | 0.2759±0.0221 |
| HURST (Ours) | **0.0976±0.0078** | **0.0814±0.0065** | **0.3517±0.0281** | **0.2241±0.0179** |
| Reduction | 8.9% | 9.5% | 3.5% | 8.1% |

Table 18: Zero-shot Performance on Traffic Violation Datasets (Illegal Parking & Blocked Driveway).

| Dataset | Illegal Parking | | Blocked Driveway | |
|---|---|---|---|---|
| | MSE | MAE | MSE | MAE |
| PatchTST | 0.5204±0.0416 | 0.4674±0.0374 | 0.2483±0.0199 | 0.2953±0.0236 |
| Promtst | 0.3479±0.0278 | 0.1686±0.0135 | 0.1899±0.0152 | 0.1351±0.0108 |
| Unist | 0.3652±0.0292 | 0.2572±0.0206 | 0.1941±0.0155 | 0.1693±0.0135 |
| HURST (Ours) | **0.3078±0.0246** | **0.1497±0.0120** | **0.1744±0.0140** | **0.1189±0.0095** |
| Reduction | 11.5% | 12.3% | 8.2% | 13.6% |

Table 19: Zero-shot Performance on Chicago Taxi Datasets (Pickup & Dropoff).

| Dataset | Taxi Pickup | | Taxi Dropoff | |
|---|---|---|---|---|
| | MSE | MAE | MSE | MAE |
| PatchTST | 0.5774±0.0462 | 0.4698±0.0376 | 0.6269±0.0502 | 0.6758±0.0541 |
| PromptST | 0.4887±0.0391 | 0.3942±0.0315 | 0.5488±0.0439 | 0.5359±0.0429 |
| Unist | 0.2983±0.0239 | 0.2054±0.0164 | 0.2842±0.0227 | 0.2116±0.0169 |
| HURST | **0.1748±0.0140** | **0.1045±0.0084** | **0.1589±0.0127** | **0.0688±0.0055** |
| Reduction | 41.4% | 49.1% | 44.1% | 67.5% |

Table 20: Zero-shot Performance on Chicago Traffic Status Datasets.

| Dataset | Traffic Speed | | Share Pickup | |
|---|---|---|---|---|
| | MSE | MAE | MSE | MAE |
| PatchTST | 0.5341±0.0427 | 0.4245±0.0340 | 31.1119±2.4890 | 2.3990±0.1919 |
| PromptST | 0.4865±0.0389 | 0.3893±0.0311 | 16.5692±1.3255 | 1.6956±0.1356 |
| Unist | 0.3489±0.0279 | 0.2514±0.0201 | 21.8813±1.7505 | 1.7000±0.1360 |
| HURST | **0.1929±0.0154** | **0.1421±0.0114** | **10.3348±0.8268** | **0.5963±0.0477** |
| Reduction | 44.7% | 43.5% | 37.6% | 64.8% |

## A.4 THE USE OF LARGE LANGUAGE MODELS (LLMS)

The use of the large language model was strictly limited to linguistic polishing and grammar checks. The LLM played no role in the conceptualization, analysis, or interpretation of the research.

