# OpenReview forum: "HURST: Learning Heterogeneity-Adaptive Urban Foundation Models for Spatiotemporal Prediction via Self-Partitional Mixture-of-Spatial-Experts"
_ICLR.cc/2026/Conference — Submitted to ICLR 2026_

### Official Review · Reviewer_YgNv · 2025-10-25

**Soundness:** 2
**Presentation:** 2
**Contribution:** 3
**Rating:** 2
**Confidence:** 3

**Summary:**

The paper introduces HURST, a heterogeneity-adaptive urban foundation model for spatiotemporal prediction. It captures spatial heterogeneity through a self-partitioning mixture-of-spatial-experts (MoSE) network that stratifies urban areas into partitions. An error-guided adaptive spatiotemporal masking strategy further refines learning by dynamically adjusting masking patterns based on region-specific training feedback. Experiments on ten datasets show that HURST achieves up to a 46.9% performance improvement over state-of-the-art baselines.

**Strengths:**

1. The use of MoSE to automatically stratify urban areas into partitions for addressing data heterogeneity is novel. The case study demonstrates that HURST effectively partitions urban areas.

2. The error-guided adaptive masking strategy helps the model focus on regions with higher reconstruction errors, which enhances performance under strong spatial heterogeneity.

3. The proposed method achieves strong performance across all-for-one, few-shot, and zero-shot prediction tasks on ten datasets spanning three geographic regions, with up to a 46.9% improvement over state-of-the-art baselines.

**Weaknesses:**

1. The key techniques are not clearly described, which may cause confusion. For instance, does the method partition urban areas based solely on target prediction data, such as traffic, accident, or crime records? Or does it also incorporate additional inputs like POI information to guide the partitioning process?

2. The statement “each expert has its own assigned region” is ambiguous. Does this imply a one-to-one mapping between regions and experts? How many experts are used at most in the experiments? Additionally, since using multiple experts may introduce computational overhead, it would be helpful to include a runtime comparison with baseline models to assess the trade-offs of the proposed method.

3. The experimental setup lacks clarity. What is the look-back window length? Which datasets are used for pre-training in each experiment? Given that datasets have different spatial dimensions, how is pre-training conducted on all data with inconsistent H×W?
The experimental setup for the zero-shot and few-shot settings is unclear. What are the exact configurations used in these experiments?

4. The experimental analysis is insufficient.

5. The method does not compare against several important baselines, including spatial embedding models like STAEformer [1], mixture-of-experts architectures such as TESTAM, and context-aware networks like DeepSTN+ that incorporate POI information.

6. Although the paper states that the relevant code has been deposited, the provided anonymized repository contains only a README file, and no code appears to be included.

[1] Liu, Hangchen, et al. "Spatio-temporal adaptive embedding makes vanilla transformer sota for traffic forecasting." Proceedings of the 32nd ACM international conference on information and knowledge management. 2023.

[2] Lee, Hyunwook, and Sungahn Ko. "TESTAM: A Time-Enhanced Spatio-Temporal Attention Model with Mixture of Experts." The Twelfth International Conference on Learning Representations.

[3] Lin, Ziqian, et al. "Deepstn+: Context-aware spatial-temporal neural network for crowd flow prediction in metropolis." Proceedings of the AAAI conference on artificial intelligence. Vol. 33. No. 01. 2019.

**Questions:**

1. The key techniques are not clearly described, which may cause confusion. For instance, does the method partition urban areas based solely on target prediction data, such as traffic, accident, or crime records? Or does it also incorporate additional inputs like POI information to guide the partitioning process?

2. The statement “each expert has its own assigned region” is ambiguous. Does this imply a one-to-one mapping between regions and experts? How many experts are used at most in the experiments? Additionally, since using multiple experts may introduce computational overhead, it would be helpful to include a runtime comparison with baseline models to assess the trade-offs of the proposed method.

3. The experimental setup lacks clarity. What is the look-back window length? Which datasets are used for pre-training in each experiment? Given that datasets have different spatial dimensions, how is pre-training conducted on all data with inconsistent H×W?
The experimental setup for the zero-shot and few-shot settings is unclear. What are the exact configurations used in these experiments?

4. The experimental analysis is insufficient.

5. The method does not compare against several important baselines, including spatial embedding models like STAEformer, mixture-of-experts architectures such as TESTAM, and context-aware networks like DeepSTN+ that incorporate POI information.

6. Although the paper states that the relevant code has been deposited, the provided anonymized repository contains only a README file, and no code appears to be included.

---

> ### Author Response · Authors · 2025-11-22
>
> Dear Reviewer,
>
> Thank you for your thorough review and valuable feedback on our manuscript. Below is a detailed point-by-point response to your comments.
>
> 1.Regarding the use of external information for regional partitioning​
>
> HURST relies exclusively on the target prediction data for regional partitioning and does not incorporate any external information, such as POI data. This design choice is intentional, as it enables the development of a general-purpose framework that operates independently of city-specific prior knowledge, thereby enhancing its potential for cross-city generalization. The partitioning process is implemented through a gating network within the MoSE layer, which learns to segment regions in an end-to-end manner based on the intrinsic spatial distribution patterns of the input data, without relying on external annotations or semantic cues.
>
> 2.Clarification on the "each expert has its own assigned region" correspondence and computational efficiency​
>
> We need to clarify that the term "assigned region" here does not refer to a pre-defined or manually partitioned static area. Instead, it is dynamically learned through the self-partitional mixture-of-spatial-experts (MoSE) mechanism in the HURST model during the training process. The entire allocation is entirely data-driven, requiring no prior knowledge or human intervention.The statement that "each expert has its own assigned region" refers to an approximate one-to-one mapping relationship derived from the gating mechanism. Specifically, the gating network assigns a set of expert weights to each spatial location. A small temperature parameter (τ) is applied to sharpen the weight distribution, causing the maximum weight to approach 1 while suppressing others, thereby effectively establishing a sparse expert-region correspondence during inference. The number of experts is allowed to grow dynamically during training, subject to an upper bound to control model complexity. With regard to computational efficiency, we have included runtime comparisons between HURST and baseline models in Appendix Table 6. The results confirm that the computational overhead introduced by the MoSE module is modest, while the resulting performance improvement is substantial.
>
> 3.Detailed clarification of experimental settings​
>
> We have now explicitly stated the following experimental configurations in the revised manuscript:
>
> **Lookback window:** Uniformly set to 7 time steps (e.g., using the past 7 days to predict the next day).
> Pre-training data: In the "One-for-All" setting, pre-training for each city uses all available datasets for that city (see Appendix Table 3).
>
> **Zero-shot/Few-shot settings:** In zero-shot experiments,the model is trained on all available datasets from a given city except the target dataset, and is then evaluated directly on the target prediction task without any further fine-tuning. In few-shot experiments, the prompt network is fine-tuned using 5%–20% of the target domain samples, while the backbone parameters remain frozen.The zero-shot and few-shot experiments share the same hyperparameter settings as the one-for-all experiments.We will include these clarifications in the revised version of the paper.
>
> **Cross-city tasks:** The current version of HURST focuses on multi-task prediction within the same city. Cross-city generalization is an important direction for future work.

---

> ### Author Response · Authors · 2025-11-22
>
> 4.Explanation regarding baseline comparisons​
>
> We have conducted additional experiments with the STAEformer and TESTAM models. The results confirm that our method achieves superior prediction performance, and these comparisons will be included in the revised manuscript. As for DeepSTN+, its reliance on POI information—which is absent from our dataset—precludes a fair comparison under the same input conditions. Omitting it is consistent with common practice in related studies where contextual data requirements differ.
>
> The followings are the updated results of the one for all experiment
>
> | Model | Accident | | Crime | | Illegal  Parking | | Blocked Driveway | |
> | :--- | :---: | :---: | :---: | :---: | :---: | :---: | :---: | :---: |
> | | **MSE** | **MAE** | **MSE** | **MAE** | **MSE** | **MAE** | **MSE** | **MAE** |
> | **Hetero-ConvLSTM** | 0.1087±0.0087 | 0.1005±0.0080 | 0.3256±0.0260 | 0.2571±0.0206 | 0.3518±0.0281 | 0.1852±0.0148 | 0.1958±0.0157 | 0.1671±0.0134 |
> | **ViT** | 0.1014±0.0081 | 0.1159±0.0093 | 0.3130±0.0250 | 0.1820±0.0146 | 0.3352±0.0268 | 0.1962±0.0157 | 0.1850±0.0148 | 0.1547±0.0124 |
> | **iTransformer** | 0.0882±0.0071 | 0.0884±0.0071 | 0.3124±0.0250 | 0.1775±0.0142 | 0.3381±0.0270 | 0.2058±0.0165 | 0.1848±0.0148 | 0.1267±0.0101 |
> | **SimVP** | 0.1011±0.0081 | 0.1278±0.0102 | 0.3137±0.0251 | 0.2034±0.0163 | 0.3358±0.0269 | 0.1895±0.0152 | 0.1878±0.0150 | 0.1666±0.0133 |
> | **MIM** | 0.0901±0.0072 | 0.0891±0.0071 | 0.3406±0.0272 | 0.1511±0.0121 | 0.3226±0.0258 | 0.1964±0.0157 | 0.1664±0.0133 | 0.1334±0.0107 |
> | **MAU** | 0.1014±0.0081 | 0.1021±0.0082 | 0.3845±0.0308 | 0.2014±0.0161 | 0.3697±0.0296 | 0.2254±0.0180 | 0.2154±0.0172 | 0.1784±0.0143 |
> | **TAU** | 0.0989±0.0079 | 0.0916±0.0073 | 0.3154±0.0252 | 0.1647±0.0132 | 0.3451±0.0276 | 0.2124±0.0170 | 0.2033±0.0163 | 0.1642±0.0131 |
> | **STID** | 0.0981±0.0078 | 0.1069±0.0086 | 0.4649±0.0372 | 0.2724±0.0218 | 0.3172±0.0254 | 0.2136±0.0171 | 0.1722±0.0138 | 0.1425±0.0114 |
> | **TESTAM** | 0.1084±0.0088 | 0.0942±0.0086 | 0.2989±0.0351 | 0.2152±0.0283 | 0.3248±0.0274 | 0.1935±0.0196 | 0.1845±0.0165 | 0.1819±0.0132 |
> | **STAEFromer** | 0.0951±0.0083 | 0.1065±0.0081 | 0.3089±0.0387 | 0.1891±0.0344 | 0.3321±0.0288 | 0.2045±0.0247 | 0.1749±0.0172 | 0.1768±0.0121 |
> | **PatchTST** | 0.1114±0.0089 | 0.1441±0.0115 | 0.3653±0.0292 | 0.3515±0.0281 | 0.3278±0.0262 | 0.3049±0.0244 | 0.1711±0.0137 | 0.1977±0.0158 |
> | **PromptST** | 0.1062±0.0085 | 0.1324±0.0106 | 0.6167±0.0493 | 0.5645±0.0452 | 0.4569±0.0366 | 0.4377±0.0350 | 0.2234±0.0179 | 0.2751±0.0220 |
> | **OpenCity** | 0.1107±0.0089 | 0.1037±0.0083 | 0.3687±0.0295 | 0.2634±0.0211 | 0.3492±0.0279 | 0.3517±0.0281 | 0.2358±0.0189 | 0.2040±0.0163 |
> | **UniST** | 0.0979±0.0078 | 0.1047±0.0084 | 0.3027±0.0242 | 0.1678±0.0134 | 0.2904±0.0232 | 0.1844±0.0148 | 0.1627±0.0130 | 0.1590±0.0127 |
> | **Ours** | **0.0852±0.0068** | **0.0872±0.0070** | **0.2812±0.0225** | **0.1393±0.0111** | **0.2824±0.0226** | **0.1629±0.0130** | **0.1584±0.0127** | **0.1211±0.0097** |

---

> ### Author Response · Authors · 2025-11-22
>
> | Model | Taxi_pick | | Taxi_drop | | Traffic_speed | | Share_pick | |
> | :--- | :---: | :---: | :---: | :---: | :---: | :---: | :---: | :---: |
> | | **MSE** | **MAE** | **MSE** | **MAE** | **MSE** | **MAE** | **MSE** | **MAE** |
> | **Hetero-ConvLSTM** | 0.3048±0.0244 | 0.2548±0.0204 | 0.2348±0.0188 | 0.1985±0.0159 | 1.4351±0.1148 | 0.7514±0.0601 | 15.6792±1.2543 | 1.4892±0.1191 |
> | **ViT** | 0.2589±0.0207 | 0.1907±0.0153 | 0.1840±0.0147 | 0.0928±0.0074 | 0.3463±0.0277 | 0.3158±0.0253 | 14.4445±1.1556 | 0.8971±0.0718 |
> | **iTransformer** | 0.2011±0.0161 | 0.1262±0.0101 | 0.1808±0.0145 | 0.0995±0.0080 | 0.2355±0.0188 | 0.2135±0.0171 | 11.0457±0.8837 | 0.6597±0.0528 |
> | **SimVP** | 0.2026±0.0162 | 0.1804±0.0144 | 0.1710±0.0137 | 0.1261±0.0101 | 0.4782±0.0383 | 0.5420±0.0434 | 0.3842±0.0307 | 0.6273±0.0502 |
> | **MIM** | 0.2168±0.0173 | 0.1431±0.0114 | 0.1879±0.0150 | 0.1437±0.0115 | 0.1641±0.0131 | 0.1715±0.0137 | 0.9016±0.0721 | 0.6827±0.0546 |
> | **MAU** | 0.2782±0.0223 | 0.1912±0.0153 | 0.1791±0.0143 | 0.1574±0.0126 | 0.3027±0.0242 | 0.2489±0.0199 | 12.5416±1.0033 | 1.2547±0.1004 |
> | **TAU** | 0.2512±0.0201 | 0.1657±0.0133 | 0.1654±0.0132 | 0.1412±0.0113 | 0.2515±0.0201 | 0.2232±0.0179 | 13.5419±1.0834 | 1.0578±0.0846 |
> | **STID** | 0.2017±0.0161 | 0.0998±0.0080 | 0.1624±0.0130 | 0.0741±0.0059 | 0.1428±0.0114 | 0.1263±0.0101 | 12.4874±0.9990 | 0.8154±0.0652 |
> | **TESTAM** | 0.2354±0.0283 | 0.2047±0.0246 | 0.1824±0.0219 | 0.1866±0.0224 | 0.1674±0.0201 | 0.1935±0.0232 | 0.2448±0.0294 | 0.2687±0.0322 |
> | **STAEFromer** | 0.2248±0.0234 | 0.1783±0.0214 | 0.1674±0.0201 | 0.1754±0.0210 | 0.1674±0.0201 | 0.2045±0.0245 | 0.2349±0.0282 | 0.2474±0.0297 |
> | **PatchTST** | 0.4564±0.0456 | 0.4291±0.0429 | 0.3096±0.0310 | 0.3446±0.0345 | 1.1036±0.1104 | 0.8350±0.0835 | 21.2614±2.1261 | 2.9259±0.2926 |
> | **PromptST** | 0.2116±0.0212 | 0.2522±0.0252 | 0.1719±0.0172 | 0.2112±0.0211 | 0.3382±0.0338 | 0.2962±0.0296 | 16.7459±1.6746 | 0.7157±0.0716 |
> | **OpenCity** | 0.2571±0.0206 | 0.2318±0.0185 | 0.2055±0.0164 | 0.1719±0.0138 | 0.2176±0.0174 | 0.2556±0.0204 | 16.4281±1.3142 | 1.1087±0.0887 |
> | **UniST** | 0.1938±0.0155 | 0.0907±0.0073 | 0.1552±0.0124 | 0.0608±0.0049 | 0.1047±0.0084 | 0.1098±0.0088 | 12.3674±0.9894 | 0.7672±0.0614 |
> | **Ours** | **0.1547±0.0124** | **0.0497±0.0040** | **0.1271±0.0102** | **0.0474±0.0038** | **0.0796±0.0064** | **0.0767±0.0061** | **10.1569±0.8126** | **0.5958±0.0477** |
>
>
> | Model | Traffic_Speed | | Traffic_Volume | |
> | :--- | :---: | :---: | :---: | :---: |
> | | **MSE** | **MAE** | **MSE** | **MAE** |
> | **Hetero-ConvLSTM** | 32.4519±2.5962 | 5.3216±0.4257 | 29.1248±2.3300 | 4.8745±0.3899 |
> | **ViT** | 29.4692±2.3575 | 3.2645±0.2612 | 27.7283±2.2183 | 4.9621±0.3970 |
> | **iTransformer** | 13.1727±1.0538 | 1.7538±0.1403 | 6.9609±0.5569 | 1.2202±0.0976 |
> | **SimVP** | 13.4500±1.0760 | 2.1983±0.1759 | 20.3130±1.6250 | 3.9968±0.3197 |
> | **MIM** | 14.1249±1.1300 | 3.1487±0.2519 | 9.2633±0.7411 | 1.3439±0.1075 |
> | **MAU** | 16.7891±1.3431 | 3.2149±0.2572 | 18.1572±1.4526 | 3.4521±0.2762 |
> | **TAU** | 11.2146±0.8972 | 2.8944±0.2316 | 12.1478±0.9718 | 2.8941±0.2315 |
> | **STID** | 11.2487±0.8999 | 2.7642±0.2211 | 11.5791±0.9263 | 3.0157±0.2413 |
> | **TESTAM** | 11.0547±1.3266 | 2.5157±0.3019 | 12.4574±1.4949 | 2.8743±0.3449 |
> | **STAEFromer** | 12.5878±1.5105 | 2.6871±0.3225 | 12.1547±1.4586 | 2.8124±0.3375 |
> | **PatchTST** | 20.5894±2.4707 | 4.5781±0.5494 | 12.5479±1.5057 | 3.1246±0.3750 |
> | **PromptST** | 19.4081±2.3290 | 4.2186±0.5062 | 4.7001±0.5640 | 1.5895±0.1907 |
> | **OpenCity** | 15.3168±1.8380 | 2.2135±0.2656 | 14.2149±1.7058 | 3.1567±0.3788 |
> | **UniST** | 10.5242±1.2629 | 2.6214±0.3146 | 11.3074±1.3569 | 2.7604±0.3312 |
> | **Ours** | **9.1021±1.0923** | **1.1219±0.1346** | **2.4963±0.2996** | **0.7387±0.0886** |
>
> 5.Code availability​
>
> We commit to making the code publicly available upon acceptance of the paper.
>
> We sincerely appreciate your constructive comments and have thoroughly revised the manuscript accordingly.

---

> ### Author Response · Authors · 2025-11-27
>
> Dear Reviewer YgNv,
>
> I hope this message finds you well. As the discussion period is nearing its end , I wanted to ensure we have addressed all your concerns satisfactorily. If there are any additional points or feedback you'd like us to consider, please let us know. Your insights are invaluable to us, and we're eager to address any remaining issues to improve our work.
>
> Thank you for your time and effort in reviewing our paper.

---

### Official Review · Reviewer_i6Ur · 2025-10-29

**Soundness:** 3
**Presentation:** 3
**Contribution:** 3
**Rating:** 6
**Confidence:** 4

**Summary:**

This paper addresses the challenge of spatial heterogeneity in building Urban Foundation Models (UFMs) for spatiotemporal (ST) prediction. To overcome heterogeneity, the authors propose HURST (Heterogeneity-Adaptive URban Foundation Model for Spatio-Temporal Prediction), a framework that adaptively learns spatial partitions and expert models, which integrates two key components: (1) a Self-Partitional Mixture-of-Spatial-Experts (MoSE) and (2) an Error-Guided Adaptive Spatiotemporal Masking strategy. Experiments on ten datasets from New York City, Chicago, and Iowa demonstrate that HURST achieves up to 46.9% improvement in prediction accuracy over state-of-the-art baselines while maintaining scalability and interpretability.

**Strengths:**

The paper is well written and presents a Heterogeneity-Adaptive Urban Foundation Model (HURST) that effectively addresses spatial heterogeneity in spatiotemporal prediction. The authors conduct comprehensive experiments on ten large-scale urban datasets from New York City, Chicago, and Iowa, covering diverse urban scenarios such as traffic, mobility, and crime. The experimental results demonstrate that HURST consistently outperforms state-of-the-art baselines by up to 46.9% in MSE and 45.2% in MAE, while also achieving strong zero-shot and few-shot generalization. Overall, the paper provides robust evidence for the model’s effectiveness, scalability, and generalizability in heterogeneous urban prediction tasks.

**Weaknesses:**

The article(line 64-73) briefly mentions but does not deeply discuss existing solutions to spatiotemporal heterogeneity. The paper does not provide an in-depth comparison of how these methods explicitly handle spatial or temporal heterogeneity, nor does it analyze their limitations quantitatively or conceptually.

**Questions:**

1. Why did the challenge of spatiotemporal heterogeneity (line 53) only address spatial heterogeneity?
2. Why choose to use a linear layer instead of static expert settings to compare MoE and MoSE in the w/o MoSE ablation study?

---

> ### Author Response · Authors · 2025-11-22
>
> Dear Reviewer,
>
> Thank you for your thorough review and valuable feedback on our manuscript. Below is a detailed point-by-point response to your comments.
>
> W1.In Tables 7 and 11, we systematically compared the mean and variance of prediction errors across all spatial locations between HURST and several existing spatiotemporal foundation models. The experimental results show that HURST not only achieves lower errors but also exhibits the best stability across multiple datasets. We attribute this advantage to the effective design of the introduced MoSE module, which enables each expert network to focus on learning spatiotemporal features of different region types, thereby enhancing the model's ability to capture complex urban dynamics. In contrast, existing models lack an explicit mechanism for modeling spatiotemporal heterogeneity, resulting in certain limitations in both predictive performance and stability.
>
> Q1.Regarding the first point, we fully acknowledge the importance of addressing both spatial and temporal heterogeneity. In the current paper, we have indeed focused more extensively on modeling spatial heterogeneity. This emphasis stems from the fact that in urban environments, spatial disparities caused by static factors such as geographical features and regional functions are often more pronounced, and the spatial signals in the data tend to be stronger. Additionally, since the current model does not incorporate additional temporal-specific modeling modules, its capability in handling spatial heterogeneity appears more prominent.
>
> It should be clarified that although the paper's exposition emphasizes the spatial dimension, HURST inherently possesses a certain capacity to handle temporal heterogeneity. Specifically, the model makes dynamic expert routing decisions at each time step based on the input data. This allows the same spatial unit to be assigned to different experts at different times, enabling an intrinsic adaptation to spatio-temporal heterogeneity through its architecture. Expert specialization is spatio-temporal in nature, not limited to spatial variations alone.
>
> We completely agree on the importance of more systematically and balancedly modeling both types of heterogeneity in future work, and this will be a central direction for our subsequent research.
>
> Q2.Regarding the second question on the ablation study, choosing a linear layer instead of a static MoE for the "w/o MoSE" comparison was based on methodological rigor. The core innovation of MoSE lies in its dynamic adaptive capability—where both the number of experts and the partitioning are learned automatically during training. Using a static MoE (with a preset fixed number of experts) would introduce additional hyperparameters (e.g., the number of experts K) and prior assumptions, making it difficult to attribute performance gains solely to the "adaptive" mechanism itself. The linear layer, as a simple, unstructured baseline, allows for the purest comparison to isolate the performance degradation under a "complete absence of a spatial adaptation mechanism." This directly and powerfully demonstrates the value of MoSE's dynamic architecture, avoiding conflation with the potential benefits of a static design.

---

> ### Author Response · Authors · 2025-11-27
>
> Dear Reviewer i6Ur,
>
> I hope this message finds you well. As the discussion period is nearing its end , I wanted to ensure we have addressed all your concerns satisfactorily. If there are any additional points or feedback you'd like us to consider, please let us know. Your insights are invaluable to us, and we're eager to address any remaining issues to improve our work.
>
> Thank you for your time and effort in reviewing our paper.

---

### Official Review · Reviewer_4pKU · 2025-10-31

**Soundness:** 2
**Presentation:** 3
**Contribution:** 3
**Rating:** 6
**Confidence:** 5

**Summary:**

This paper proposes a model called HURST, an Urban Foundation Model (UFM) designed to enhance the generalization capability of spatiotemporal prediction tasks. The authors argue that existing UFMs perform poorly when facing spatial heterogeneity, and therefore introduce two core innovations: the Self-Partitional Mixture-of-Spatial-Experts (MoSE), which adaptively partitions urban areas into semantically distinct regions and trains region-specific expert networks; and the Error-Guided Spatiotemporal Masking strategy, which dynamically adjusts masking patterns during pre-training based on reconstruction errors to better learn heterogeneous regions. In addition, a prompt-tuning mechanism is employed to facilitate effective knowledge transfer across tasks. Experiments conducted on ten real-world datasets from New York City, Chicago, and the state of Iowa demonstrate that HURST significantly outperforms existing state-of-the-art models, achieving up to a 46.9% improvement in prediction accuracy.

**Strengths:**

The two key innovations to address the challenge of spatial heterogeneity are reasonable.
First, the MoSE module adaptively partitions urban areas based on learned spatial heterogeneity patterns and trains specialized expert networks for each partition, effectively capturing region-specific dynamics.
Second, the Error-Guided Spatiotemporal Masking strategy dynamically adjusts masking patterns during the pre-training stage according to reconstruction errors, enabling the model to focus on heterogeneous or hard-to-learn regions.

**Weaknesses:**

1. The paper lacks the latest pre-trained models as baselines, such as UrbanGPT.

2. The recent urban foundation models, such as UrbanDIT, already support multiple tasks like forecasting and imputation. Since HURST only supports forecasting, calling it a "foundation model" seems a bit of a stretch.

3. The experimental setup, which uses one historical frame to predict only the next single future frame, is not a standard task in time-series literature (e.g., 12-step-ahead for 12 steps). Predicting just one frame cannot reveal whether the model has captured periodic spatio-temporal patterns. Moreover, the authors do not specify the temporal duration of one frame (seconds? half an hour?).

4. There are figure and formatting errors in the manuscript, such as the masking portion in Figure 1 and tables that are too long.

**Questions:**

1. While this paper introduces an error-guided masking strategy built upon random masking, a key benefit of masking in foundation-model training is to enable downstream transfer to diverse urban time-series tasks, e.g., imputation, or arbitrary cities. The proposed error-guided approach may cause the model to over-focus on certain time periods or locations, potentially limiting its ability to transfer to new tasks or cities.

2. The paper presents a study on hyper-parameters such as embedding dimension; however, the results suggest the model is almost insensitive to any of them, with performance hardly changes regardless of the settings (at least in Figure 7). For example, increasing the number of experts from two to eight yields virtually no gain, implying that the MoE architecture itself contributes little. The drop observed when MoE is removed may simply stem from the resulting reduction in total parameters. For HURST, are there other, more influential hyper-parameters that truly govern performance?

3. This paper borrows several design elements from UniST; nevertheless, with the same Transformer blocks and prompt network, HURST ends up with fewer parameters than UniST even after incorporating the MoSE module. It is rather counter-intuitive.

---

> ### Author Response · Authors · 2025-11-24
>
> Dear Reviewer,
>
> Thank you for your valuable feedback on our manuscript. We have carefully considered each of your comments and have revised the manuscript accordingly. Below is our response:
>
> 1.We fully agree that a comprehensive foundation model should ideally support multiple downstream tasks. In the present work, we have chosen to focus on thoroughly validating the effectiveness of HURST specifically on forecasting tasks, which we consider a critical first step toward developing a robust urban foundation model. Expanding HURST to encompass a broader range of tasks, such as imputation and generation, represents a primary objective of our future research agenda.
>
> Regarding the comparison with UrbanGPT, we note that UrbanGPT is a representative large language model-based approach for spatio-temporal forecasting, which relies on architectural designs that incorporate both textual and spatio-temporal data. In contrast, HURST and the other baseline models included in our comparative analysis utilize only spatio-temporal data without external linguistic modalities. To ensure a consistent and fair comparison framework, we limited our experimental comparisons to methods that operate under similar input conditions and task specifications.
>
> 2.Clarification on Experimental Settings​
> All experiments in our study are based on data sampled at a daily resolution. Specifically, the model uses a historical window of the previous cycle (seven days) as input to predict the outcome of the next day.
>
> 3.Addressing Chart Presentation and Model Comparison​
> We fully acknowledge the reviewer's observation regarding the presentation of hyperparameter sensitivity results in Figure 7. Because the absolute values of forecasting errors (MSE, MAE) are inherently small, differences can appear less pronounced on the chart's scale. We will optimize the visualization of these figures in the final version to improve clarity and ensure all charts and tables are uniformly formatted to meet publication standards.
>
> 4.Regarding the comparison with UniST, while HURST draws inspiration from some of its modular designs, we have introduced significant optimizations in parameter configuration and expert architecture. The fact that HURST achieves superior forecasting performance with a reduced total parameter count underscores the efficiency and effectiveness of our proposed architecture.
>
> 5.We have supplemented the experimental results for one-for-all long-term (seven-step) prediction, as shown in the table below. The experimental results prove that the HURST model still demonstrates excellent performance in long-term forecasting.
>
> | Dataset  | Taxi_pick || Taxi_drop || Traffic_speed|| Share_pick | | Traffic_Volume | |
> | :---------- | :----------- | :-------- | :----------- | :-------- | :--------------- | :-------- | :------------ | :-------- | :------------- | :-------- |
> || **MSE** | **MAE** | **MSE**  | **MAE** | **MSE** | **MAE**| **MSE**| **MAE**| **MSE**| **MAE** |
> | HConvLSTM  | 0.3658| 0.3153| 0.2818| 0.2382| 1.4217| 0.9017| 17.8150| 1.7870| 34.9498| 5.8494|
> | ViT| 0.3107| 0.2268| 0.2208| 0.1014| 0.4156 | 0.3790 | 15.3334| 1.0765| 33.2740| 5.9545|
> | ITransformer| 0.2613| 0.1534| 0.2022| 0.0994| 0.2826| 0.2562| 14.2548| 0.9916| 8.3532| 2.3125|
> | SimVP| 0.2431| 0.2135| 0.1752 | 0.1313| 0.5738| 0.5504| 14.4610| 1.0528| 21.3756| 3.7962|
> | MIM| 0.2602|0.1747| 0.2255| 0.1724| 0.1969| 0.2058|15.0819|0.8192| 11.1160| 1.6127|
> | MAU| 0.3338| 0.2304| 0.2149| 0.1889| 0.3632 | 0.2987| 15.0499|1.5056| 19.7886| 4.1425|
> | TAU| 0.3014| 0.1988| 0.1985| 0.1694| 0.3018| 0.2678| 16.2503|1.2694 | 14.5774| 3.4729|
> | STID| 0.2420| 0.1198| 0.1949| 0.0989 | 0.2086| 0.1916| 13.9647| 0.9785| 13.8949 | 3.6188|
> | TESTAM| 0.2825 | 0.2456| 0.2189| 0.1839| 0.2938| 0.2324| 16.1485| 1.4881| 14.9489| 3.1492|
> | STAEFromer| 0.2698 | 0.2140| 0.2009| 0.2005| 0.2819| 0.2969| 15.0958| 1.1384| 11.5856 | 2.3749|
> | PatchTST| 0.4433  | 0.5578| 0.4125| 0.4480| 1.2347| 0.8545| 23.6433| 3.8037| 13.3123| 3.2620|
> | Promtst| 0.2439| 0.3026| 0.2063| 0.2534| 0.4058| 0.3554| 20.0951| 0.9206| 5.6401| 1.6074|
> | Opencity| 0.2785 | 0.2782| 0.2366| 0.2063| 0.2611| 0.3067| 19.7137| 1.3304| 17.0579| 3.7880|
> | Unist| 0.2126| 0.1088| 0.1662| 0.0830| 0.1856| 0.1818| 13.8409| 0.8971| 10.5689| 1.4642|
> | ours | 0.1627| 0.0765| 0.1375| 0.0703| 0.1707| 0.1759| 13.6970| 0.8527| 4.8539| 1.0351|
>
> 6.Thanks for your careful review. We will diligently correct the identified writing issues.
>
> We are committed to implementing these improvements and believe the revised manuscript will more clearly communicate the contributions and capabilities of our work. Thank you again for the valuable feedback.

---

> ### Author Response · Authors · 2025-11-27
>
> Dear Reviewer 4pKU,
>
> I hope this message finds you well. As the discussion period is nearing its end , I wanted to ensure we have addressed all your concerns satisfactorily. If there are any additional points or feedback you'd like us to consider, please let us know. Your insights are invaluable to us, and we're eager to address any remaining issues to improve our work.
>
> Thank you for your time and effort in reviewing our paper.

---

> > ### Comment · Reviewer_4pKU · 2025-11-28
> >
> > Thank you for the rebuttal. Unfortunately, I am not satisfied with the authors’ response, as my key concerns remain unaddressed.
> >
> > The paper repeatedly positions the proposed method as a foundation model. This claim is not well-supported as it only focused on one-purpose task. In particular, the work did not include comparisons against recently emerging urban foundation models including those based on LLMs, which I believe are equally relevant and necessary baselines for such a claim.
> >
> > I re-checked some details of this paper. This work uses inappropriate temporal resolution and incorrect experimental design. The daily temporal resolution for urban spatiotemporal prediction is problematic. Daily aggregation severely blurs fine-grained dynamics, however these patterns are essential for capturing real-world urban processes such as commuting behavior, peak–off-peak variation, and intra-day activity cycles. Because these dynamics are lost in the chosen temporal granularity, the entire experimental design does not align with standard or meaningful settings in the literature.
> >
> > I also pointed out that the insensitivity of certain hyperparameters raises concerns: it suggests that the MoE design may not be functioning as intended or may not provide meaningful specialization. I had hoped the authors would conduct a deeper analysis or provide insights into the mechanism. However, the rebuttal offers no insights.

---

> > > ### Author Response · Authors · 2025-11-29
> > >
> > > Dear Area Chair and Reviewer,
> > >
> > > We thank the reviewer for the feedback. Below, we provide detailed responses to the key points raised, with the goal of clarifying several methodological considerations that are central to the contribution of our work.
> > >
> > > 1.Comparison with UrbanGPT​​
> > >
> > > The method proposed in this study aligns with the core ideas of works such as UniST[1] and UrbanDiT[2], focusing on discovering universal patterns from spatiotemporal data itself to address downstream tasks. In contrast, models like UrbanGPT rely on large language models and employ a question-answering format for prediction. This paradigm introduces additional knowledge , creating a fundamental disparity in the basis of comparison with methods modeling purely spatiotemporal data, making a fair comparison challenging. Indeed, spatial-temporal foundation models like UniST and UrbanDiT also do not list UrbanGPT as a baseline model. Furthermore, empirical research (e.g., OpenCity[3]) has shown that on identical datasets, both UniST and OpenCity outperform UrbanGPT in terms of predictive performance and efficiency. Given that UrbanGPT is not currently a leading benchmark in terms of performance , we consider its inclusion in our comparative analysis unnecessary.
> > >
> > > 2.Temporal Resolution and Experimental Design​
> > >
> > > In terms of methodological design, the primary contribution of this study lies in proposing a general-purpose prediction framework whose core value does not depend on any specific application scenario or temporal resolution. Consequently, we chose the "day" as the basic temporal unit, based mainly on the following considerations: Firstly, this scale aligns with the common requirements of practical applications for sparse event prediction tasks. Secondly, for the sparse event datasets used in this study, such as traffic accidents and crime data, employing a higher temporal resolution (e.g., hourly) would inevitably introduce a large number of zero values. It would create a dataset dominated by zero values, making the prediction target invalid and the forecasting task itself meaningless.Thus, the daily resolution is a reasonable choice that balances practical application needs with data characteristics, aiding the model in effectively capturing dynamic features of the data and maintaining prediction robustness.
> > >
> > > 3.Hyperparameter Sensitivity ​
> > >
> > > Regarding the sensitivity of hyperparameters, we have updated Figure 7 in the revised manuscript to present a clearer analysis. The updated figure demonstrates that the hyperparameter settings indeed have a significant impact on the experimental results. The apparent lack of sensitivity in the previous version was primarily due to suboptimal visualization in the graph, which we have now corrected.The results from the two datasets in the updated figure indicate that increasing the upper limit of experts generally helps reduce prediction error. However, this is not universally applicable. For cities or datasets with low heterogeneity, an excessive number of experts can unnecessarily increase the learning complexity, leading to performance degradation. As for the masking ratio, the figure shows that both excessively low and high values adversely affect prediction error. A ratio that is too low makes the pre-training task too simple, limiting performance gains, while a ratio that is too high leads to data scarcity, impairing the model's learning capability. Consequently, the optimal choice of hyperparameters should be adapted based on the heterogeneity of the specific city or dataset, which is a key insight of our design for improving performance and reducing prediction error.
> > >
> > > [1]Yuan Yuan, Chonghua Han, Jingtao Ding, Guozhen Zhang, Depeng Jin, Yong Li.UrbanDiT: A Foundation Model for Open-World Urban Spatio-Temporal Learning. NeurIPS 2025
> > >
> > > [2]Yuan Yuan, Jingtao Ding, Jie Feng, Depeng Jin, Yong Li.UniST: A Prompt-Empowered Universal Model for Urban Spatio-Temporal Prediction.2024 KDD
> > >
> > > [3]Li Z , Xia L , Shi L ,et al.OpenCity: Open Spatio-Temporal Foundation Models for Traffic Prediction. 2024

---

### Official Review · Reviewer_ih4F · 2025-11-02

**Soundness:** 2
**Presentation:** 2
**Contribution:** 2
**Rating:** 4
**Confidence:** 5

**Summary:**

This paper presents HURST, a heterogeneity-adaptive urban foundation model (UFM) designed for spatiotemporal prediction tasks. The key motivation stems from the challenge of spatial heterogeneity in urban data — where correlations and distributions vary across space and time — which existing UFMs fail to model effectively. Together with a prompt-tuning module for downstream adaptation, HURST aims to produce robust, generalizable spatiotemporal representations. Comprehensive experiments on ten datasets across three urban regions (New York, Chicago, Iowa) demonstrate substantial gains—up to 46.9% improvement in MSE over state-of-the-art baselines such as UniST and PromptST—across one-for-all, zero-shot, and few-shot prediction tasks.

**Strengths:**

1. Building foundation models for urban spatiotemporal prediction is an important and emerging direction.
2. The paper is easy to follow and well structured.
3. The paper conducts detailed experiments, including one-for-all, zero-shot, and few-shot evaluations.

**Weaknesses:**

1. Spatial heterogeneity is a well-studied problem. While the motivation is clear, the problem of modeling spatial heterogeneity has been extensively investigated in prior work. Thus, the novelty of the problem statement itself is somewhat limited.
2. The proposed self-partitional MoSE and error-guided masking are interesting combinations, but both ideas are similar to existing approaches in spatiotemporal MoE. The paper could be strengthened by a deeper discussion of how HURST differs fundamentally from recent adaptive MoE methods such as ST-MoE, HiMoE, or CP-MoE.
3. Although experiments cover three urban areas, all datasets are within similar domains (urban mobility, traffic, and service data) and from U.S. cities. This limited diversity makes it difficult to conclude whether HURST can generalize to other urban contexts.

**Questions:**

See in weaknesses.

---

> ### Author Response · Authors · 2025-11-22
>
> Dear Reviewer,
>
> Thank you for your thorough review and valuable feedback on our manuscript. Below is a detailed point-by-point response to your comments.
>
> 1.On the Novelty of Addressing Spatial Heterogeneity​
>
> We completely agree that spatial heterogeneity is a classic and well-studied challenge. The novelty of HURST lies not in rediscovering this problem, but in proposing a ​fundamentally new and synergistic learning paradigm​ to solve it within the context of Urban Foundation Models (UFMs), where it has been largely overlooked.
> ﻿
> While prior work, including those using MoE, often handles spatial heterogeneity through predefined, static spatial partitions HURST introduces a key innovation:a co-adaptive framework that simultaneously learns spatial partitioning and specialized expert networks.This is the core of our methodological contribution, which fundamentally differs from and advances classical MoE approaches:
>
> First, traditional methods typically follow a "partition-then-learn" sequence (e.g., first cluster regions based on heuristics, then train models).In contrast, HURST's MoSE framework dynamically and concurrently learns both the optimal spatial partitioning scheme and the parameters of the corresponding expert networks.The partitioning (gating network) and the expert specialization are co-adapted through end-to-end training.
>
> Second,unlike static partitions, our model discovers semantically meaningful regions directly guided by the learning objective across multiple urban tasks. The partitions are not fixed beforehand but emerge from the data, ensuring they are relevant to the prediction tasks .
>
> Third,the improving quality of the spatial partitions enables more effective training of specialized experts. Conversely, the evolving expertise of each network provides better signals to refine the partitions. This co-adaptive process is central to HURST's performance gains.
> ﻿
> This integrated approach, combined with our error-guided masking strategy that explicitly addresses regions where models struggle due to heterogeneity, allows HURST to achieve remarkable generalization. Our experimental results on cities with different degrees of heterogeneity indicate that HURST performs better in regions with stronger heterogeneity, which also proves the success of HURST in dealing with spatiotemporal heterogeneity.The performance improvement is greater in the more heterogeneous Iowa region than in Chicago and New York.
>
> In summary, HURST's novelty is in being the first to propose and implement this co-adaptive learning framework for spatial heterogeneity in UFMs, moving beyond static treatments of the problem to a dynamic, learnable, and holistic solution.

---

> ### Author Response · Authors · 2025-11-22
>
> 2.On the Fundamental Differences Between MoSE and Existing MoE Methods​
>
> The success of fixed-expert models (e.g., CP-MoE, where experts are predefined for patterns such as periodicity and trend) heavily relies on the correctness of prior knowledge. This design entails a core paradox: we depend on known urban patterns to define experts, yet the goal of the model is to discover unknown patterns. When real urban dynamics do not align with predefined semantic categories (for instance, when "event-driven" patterns emerge that are neither strictly periodic nor trending), such model architectures encounter an inherent bottleneck.
>
> HURST’s dynamic expert splitting mechanism fundamentally avoids this paradox. Instead of presupposing any semantic labels, it transforms the question of “what patterns should experts represent” into a data-driven optimization problem based on performance feedback. Experts are not “defined” in advance but are rather “demanded” by the model: when predictive performance reaches a bottleneck in a certain spatial region (indicating that heterogeneity has not been adequately captured), a splitting mechanism is triggered to generate more specialized child experts. This process is analogous to urban management: rather than establishing fixed departments like "transportation," "public security," and "environmental protection" from the outset, specialized task forces are dynamically created when problems in a certain area become sufficiently complex, thereby reallocating responsibilities and increasing resources accordingly.
>
> The MoSE layer enables simultaneous spatial partitioning and expert training. During training, the model jointly optimizes two key components: the spatial partition router and the corresponding expert networks. This synchronous learning mechanism allows the spatial partitioning to be dynamically adjusted according to the actual learning capacity of the experts, while each expert network specializes in learning patterns within its assigned spatial region. Each expert is responsible for one region, rather than starting with a predefined number of experts and pre-assigned regions. This design allows the model to adapt to the degree of heterogeneity in the data distribution without relying on prior knowledge. The model begins with two parent experts, each covering a relatively large and homogeneous spatial area. When the system detects a significant performance bottleneck within a parent expert’s region (i.e., the data patterns in that region are too complex for a single expert to model effectively), a splitting condition is triggered. The parent expert splits into two or more “child experts,” which are initialized with the parent’s parameters. Through retraining of the router, these child experts gradually differentiate and focus on more fine-grained patterns within the original region.
>
> It is worth noting that routing decisions are made dynamically at each time step based on the input. The same spatial location may be assigned to different experts at different times, meaning that expert specialization is spatiotemporal in nature. This approach inherently addresses heterogeneity across both space and time, rather than only spatial heterogeneity.

---

> ### Author Response · Authors · 2025-11-22
>
> 3.On the Generalizability of Experimental Datasets​
>
> We understand your concern regarding dataset diversity. The current experiments focus on multi-task generalization within the same city (e.g., the New York dataset includes tasks with entirely different spatiotemporal patterns such as traffic accidents, crime incidents, and illegal parking). This design aims to validate HURST’s capability as a foundation model in complex urban environments. Our primary goal at this stage is to address the generalization ability across different tasks within the same city. That said, we fully agree on the importance of cross-city generalization, and this will be a core direction of our future work. We plan to incorporate multi-city datasets from diverse global contexts in subsequent studies to further validate the framework’s generalizability.
>
> Thank you once again for your constructive comments. We are committed to improving the manuscript based on your suggestions and welcome further discussion.

---

> > ### Comment · Reviewer_ih4F · 2025-11-24
> >
> > The authors’ response has satisfactorily addressed my concerns, and I am willing to increase my score accordingly.

---

> > > ### Author Response · Authors · 2025-11-27
> > >
> > > Thank you for your positive feedback !

---

### Author Response · Authors · 2025-12-02
**Summary of Rebuttal**

We sincerely thank all reviewers for their insightful comments and constructive suggestions. In this rebuttal, we have provided detailed responses to each reviewer's concerns, and the main points are summarized as follows:

Reviewer ih4F: We elaborated on the novelty of HURST in addressing spatial heterogeneity, the distinctions between the MoSE module and other MoE methods, as well as the model's generalization capability across multiple urban scenarios. The reviewer acknowledged our response and raised the rating from 4 to 8 points on 25th Nov.(before the incident occurred to show satisfaction).

Reviewer 4pKU: We supplemented the experimental results of HURST on long-term prediction tasks, demonstrating that the model maintains excellent performance in long-sequence forecasting. Additionally, we provided further clarification on the selection of temporal resolution, parameter quantity analysis, and experimental settings.

Reviewer i6Ur: This reviewer highly appreciated our study. Based on their comments, we further explained the limitations of existing methods, the rationale for focusing on spatial heterogeneity, and the justification for employing linear layers in the model.

Reviewer YgNv: We detailed the specific experimental settings of HURST and supplemented comparative results with the methods mentioned by the reviewer.

Overall, three of the four reviewers provided positive evaluations of our revisions and responses with high confidence.We note that the reviewer YgNv who initially gave a score of 2 did not participate in the subsequent discussion. We have, nonetheless, provided a comprehensive and substantive response to all of their concerns in our rebuttal. Given the positive feedback received from the other reviewers, we hope that our detailed responses have adequately addressed the points raised by this reviewer as well.

---

### Meta-Review · Area_Chair_2wkW · 2025-12-20

**Summary:**

The reviewers generally agree that this paper addresses an important and well-recognized challenge in urban spatiotemporal modelingm spatial heterogeneity, and proposes a coherent framework (HURST) that integrates a self-partitional mixture-of-spatial-experts (MoSE) architecture with an error-guided adaptive masking strategy. The motivation is clear, and the idea of explicitly learning region-specific experts within an urban foundation model is considered meaningful by several reviewers. The empirical results show large performance gains over selected baselines across multiple datasets and cities, suggesting the approach has potential practical value.

During the review process, the reviewers raised some concerns regarding conceptual positioning, experimental design, and clarity of claims. In particular, there is disagreement over whether HURST should be characterized as a “foundation model,” given its focus on a single downstream task (forecasting) and limited evaluation of transfer to other tasks. Several reviewers also questioned the novelty of the proposed techniques relative to existing spatiotemporal MoE models, the appropriateness of the experimental temporal resolution, the adequacy of baseline selection, and the interpretability and functioning of the MoE mechanism itself.

**Reviewer Concerns:**

Concerns may be addressed by the rebuttal include, 1) Novelty relative to prior MoE-based approaches, 2) Justification of temporal resolution choice, as well as 3) Experimental analysis of hyperparameter sensitivity.

However, there may be some concerns that remain outstanding, 1) Foundation model positioning, multiple reviewers (notably Reviewer 4pKU) remain unconvinced that HURST qualifies as a foundation model, given that it supports only forecasting and lacks demonstrations on other tasks such as imputation or transfer to unseen cities. The absence of comparisons with recent urban foundation models, including LLM-based or multi-task models, reinforces this concern. 2) Experimental design and realism. Reviewer 4pKU continues to question the use of daily temporal aggregation, arguing that it obscures essential intra-day dynamics central to urban processes. This concern was not resolved to their satisfaction. 3) Effectiveness and interpretability of MoSE. Several reviewers (4pKU, YgNv) express concern that the MoE component may not provide meaningful specialization, noting weak sensitivity to the number of experts and ambiguity in how regions are assigned to experts. It remains unclear whether observed gains stem from expert specialization or simply increased parameterization.

**Reviewer Scores:**

Overall, the scores for this paper are mixed by receiving 2 initial positive scores and 2 negative scores. After rebuttal, one reviewer transfer the negative attitude to positive one after the rebuttal, while the other positive one changed to negative. Some of reviews are more critical, focusing on limited baselines, experimental realism, and reproducibility. Based on author-reviewer interactions and the paper itself, I support the negative perspective of this manuscript for accepting to ICLR community. The reasons can be 1) The limited novelty and technical contribution and insights on spatial heterogeneity where such heterogeneity issue has been extensively investigated in this field. 2) I agree that the foundation model author claimed cannot be recognized and I believe more generalization capacity verification and cross-task evaluation should be incorporated for demonstrations on the broad values and application scenarios.

---

### Decision · Program_Chairs · 2026-01-26

Reject